# miR-133a-3p and miR-338-3p shape neural crest derivatives in zebrafish

Tomás J Steeman[1], Mercedes Torres[1], Andrea MJ Weiner[1,2], Juan A Rubiolo[2,3], Laura E Sánchez[2], Gabriela Coux[1], Nora B Calcaterra[1]

Neural crest cells are a transient, multipotent population that gives rise to diverse structures during vertebrate embryonic development, including craniofacial cartilage and pigment cells. Although the transcriptional regulation of neural crest development is well characterized, the role of microRNAs remains less understood. Using a double-transgenic zebrafish model expressing fluorescent reporters under the *sox10* promoter, combined with fluorescence-activated cell sorting and RNA sequencing, we identified microRNAs enriched in neural crest cells. We focused on miR-133a-3p and miR-338-3p, previously linked to tumor suppression, to explore their developmental roles. The overexpression of either microRNA led to craniofacial cartilage malformations and reduced melanophore number, accompanied by decreased expression of key regulators including *sox9b*, *sox10*, and *runx3*. Reporter assays confirmed direct targeting of the *sox9b* 3′untranslated region. In addition, miR-338-3p overexpression increased neural crest cell numbers, suggesting a role in proliferation. These findings uncover novel functions for miR-133a-3p and miR-338-3p in vertebrate craniofacial and pigment cell development, highlighting shared regulatory features between embryogenesis and tumorigenesis.

## Introduction

During vertebrate embryogenesis, the development of organs depends on the coordinated formation, remodeling, and, in some cases, the regression of various tissues and structures. Among these, the neural crest (NC) is a transient, multipotent cell population that emerges early in development and plays a pivotal role in generating diverse cell lineages. Initially located dorsally to the neural tube, NC cells (NCC) undergo extensive proliferation followed by epithelial-to-mesenchymal transition (EMT), which facilitates their delamination and migration (Thiery et al, 2009; Theveneau & Mayor, 2012).

Subsequently, these cells differentiate into a variety of derivatives, including neurons, glial and Schwann cells, craniofacial chondrocytes, pigment cells, and endocrine cells (Dupin & Le Douarin, 2014). Defects in NC development are linked to neurocristopathies and certain cancers. Thus, elucidating the molecular mechanisms governing NC formation, migration, and differentiation is critical for understanding the etiology of these disorders and for advancing therapeutic approaches (Vega-Lopez et al, 2018; Weiner et al, 2020). NC development is regulated by a complex network of transcription factors (TFs) and signaling molecules, organized in a gene regulatory network (GRN) (Martik & Bronner, 2017) that operates in a spatiotemporal manner. In zebrafish, NC formation is initially guided by extrinsic factors, such as the BMP and WNT pathways (Hammerschmidt et al, 1996; Schumacher et al, 2011), which activate NC-specific genes such as *foxd3*, *snai1/2*, and *sox10* (Lewis et al, 2004). During EMT, the expression of new TFs such as *twist1a/b* facilitates the switch from epithelial to mesenchymal cadherins, among other changes, enabling migration (Cano et al, 2000; Ferronha et al, 2013; Martik & Bronner, 2017). Once NCC reach their destination, additional external signals induce further regulatory changes specific to the GRN of each derivative.

NCC give rise to pigment cells, such as melanophores, iridophores, and xanthophores in zebrafish. Melanophores are responsible for black pigmentation and emerge at 24 hours post-fertilization (hpf); their appearance depends on the expression of *sox10* and *pax3/7*, which activate the master gene *mitfa* (Kelsh et al, 1996). Mitfa, in turn, induces the expression of melanogenic genes such as *dct*, *tyr*, and *pmel* (Bharti et al, 2006; Greenhill et al, 2011). Epigenetic regulators like HDAC1 appear to repress alternative NC fates (Greenhill et al, 2011), favoring melanophore differentiation. Iridophores, responsible for iridescence through guanine crystals, appear at around 72 hpf (Kelsh et al, 1996; Gur et al, 2020) and follow a similar regulatory pathway but diverge through Tfec-mediated activation of TFs *ltk* and *sox10* (Lopes et al, 2008; Kaller & Hermeking, 2016; Petratou et al, 2018), among other factors. NCC also contribute to craniofacial chondrocyte development (Kimmel et al, 2001; Glasauer & Neuhauss, 2014) and skull formation in zebrafish. In this process, *sox9* paralogs (*sox9a* and *sox9b*) (Chiang et al, 2001; Yan et al, 2005), modulated by *runx3* and *egr3*, regulate

[1]Laboratorio de Biología Celular y Molecular del Desarrollo, Instituto de Biología Molecular y Celular de Rosario, CONICET-UNR, Rosario, Argentina   [2]Departamento de Zoología Genética y Antropología Física, Facultad de Veterinaria, Universidade de Santiago de Compostela, Lugo, Spain   [3]Laboratorio de Biotecnología Acuática, Facultad de Ciencias Bioquímicas y Farmacéuticas, Universidad Nacional de Rosario, Rosario, Argentina

Correspondence: steeman@ibr-conicet.gov.ar

chondrocyte stacking and proliferation (Dalcq et al, 2012). Sox9a/b, in conjunction with Sox5/6, activate the expression of *col2a1a*, a key collagen-coding gene in cartilage, and *agc1*, a marker of cartilage differentiation. Given the complexity of craniofacial development, disruptions in these regulatory pathways can contribute to congenital disorders.

Recent studies have highlighted the critical role of microRNAs (miRNAs) in NC development (Wienholds et al, 2003; Weiner, 2018). These 22-nt endogenous, noncoding RNAs regulate mRNA stability and transcription (Kloosterman et al, 2006) through the miRISC, which includes the RNase Argonaute. The miRISC targets mRNA through the binding of a specific "seed region" of the miRNAs to complementary sites within the mRNA (Lewis et al, 2005; Bartel, 2009), typically within the mRNA's 3'UTR, and each miRNA family has an average of 300 binding sites under selective pressure (Stark et al, 2005; Friedman et al, 2009). As a result, a single mRNA can be regulated by multiple miRNAs, whereas each miRNA can regulate numerous genes. MicroRNAs are indispensable in early embryonic development, influencing critical events such as brain morphogenesis and maternal mRNA clearance (Giraldez et al, 2005, 2006). In the context of NC development, miRNA dysregulation has been implicated in neuro-cristopathies, including DiGeorge syndrome and congenital central hypoventilation syndrome (Wienholds et al, 2003; Antonaci & Wheeler, 2022; Bamforth & Burn, 2023). Furthermore, miRNAs are often dysregulated in tumors, acting as either oncogenic (*oncomiRs*) or tumor-suppressive (*anti-oncomiRs*) regulators (Svoronos et al, 2016; Syeda et al, 2020). NC-derived cancers, such as neuroblastomas and melanomas, share molecular features with developmental processes, highlighting the need for further exploration into the roles of miRNAs in both NC development and cancer progression (Antonaci & Wheeler, 2022).

In this study, we aim to identify and characterize miRNAs implicated in NC development through gain-of-function experiments in zebrafish embryos. By leveraging a double-transgenic zebrafish model (Tg(*sox10*:eGFP, *sox10*:mRFP)), we isolated NC and non-NC populations via FACS and conducted transcriptomic analyses. Our findings reveal that some specific miRNAs regulate key genes involved in NC-derived structures, particularly craniofacial cartilage and melanophores, through interactions with transcription factors such as Sox9b and Runx3. Notably, these miRNAs share regulatory mechanisms with tumor suppressor pathways, suggesting a broader role in cellular plasticity and differentiation. This research advances our understanding of vertebrate embryogenesis and provides valuable insights into the molecular mechanisms underlying neurocristopathies, NC-derived cancers, and tumorigenesis, underscoring the need for further investigation into the regulatory functions of miRNAs in both developmental and pathological contexts.

# Results

## miR-133a, miR-199, and miR-338 are overrepresented in NCC

Using the double-transgenic zebrafish line Tg(*sox10*:eGFP, *sox10*:mRFP) (Kwak et al, 2013), we separated the NCC and non-NCC

populations with FACS from 16-hpf (before the NCC migration and differentiation of NC derivatives) and 28-hpf (after NCC migration and during differentiation of derivatives) embryos and performed RNA-seq on each subpopulation.

At 16 hpf, we identified a total of 225 miRNAs, with 60 and 57 being miRNAs specific to NCC and non-NCC populations, respectively (Fig 1A x-axis; fold enrichment > 2, Table S1, Fig S1B). In the NCC subpopulation, we found several previously reported NCC-associated miRNAs, such as miR-96 and miR-204 (Gessert et al, 2010; Avellino et al, 2013). At 28 hpf, we identified 50 miRNAs in NCC, of which 12 miRNAs were overrepresented compared with non-NCC and one was uniquely present in this subpopulation (miR-24, Fig 1A y-axis, Fig S1C). Conversely, the non-NCC population at 28 hpf contained a total of 187 miRNAs, with 138 unique to this subpopulation and 17 overrepresented compared with NCC. Only three miRNAs were overrepresented in both time points of the NCC (miR-125-5p, miR-16b, and miR-462), previously reported in neural or neural crest development, immune response, and hematopoiesis (Boissart et al, 2012; Singh et al, 2015; Tian et al, 2016; Huang et al, 2019; Antonaci & Wheeler, 2022) (circled in Fig 1A, Table S1).

Further analysis was performed on miRNAs overrepresented in NCC at the 16-hpf stage in zebrafish, before NCC differentiation and migration. We focused on this early time point to identify miRNAs involved in the initial specification of NCC. Based on bioinformatics analyses including STRING protein–protein interaction networks, Gene Ontology (GO) classifications, and KEGG pathway enrichments, we identified miR-133a-3p, miR-199-3p, and miR-338-3p as key candidates for further investigation. These candidates were prioritized because they had been reported as tumor suppressors in cancer models, where they regulate EMT, cell migration, and proliferation, processes analogous to those in NCC development (summarized in Table S2). Among these candidates, miR-199-3p and miR-338-3p were absent, whereas miR-133a-3p was underrepresented in the 28-hpf NCC population (Figs 1A and S1B and C).

Stem–loop RT–qPCR was performed on total RNA extracted from freshly dissociated, fluorescently sorted cells of Tg(*sox10*:eGFP, *sox10*:mRFP) (Kwak et al, 2013) embryos at 16 hpf. miRNAs miR-133a-3p, miR-199-3p, and miR-338-3p were found to be overexpressed in NCC (Fig 1B), consistent with RNA-seq data. As a non-NCC miRNA control, we analyzed miR-145-5p expression (Fig 1B), showing similar expression levels in NCC and non-NCC populations, as observed in the RNA-seq results (Fig 1A, black dot).

## In silico target prediction and functional enrichment analysis

To identify potential targets of the candidate miRNAs in NCC development, we performed computational predictions followed by functional enrichment analysis. Results are briefly described below.

### miR-133a-3p

We retrieved a total of 2,966 genes containing miR-133a-3p putative binding sites from the TargetScan database (context+ score between −0.01 and −1.05, Table S3). According to g:Profiler ($P_{adj}$ < 0.05, g:SCS threshold), multiple terms associated with neuronal development were overrepresented (Fig 2A, Table S4). Additional terms, such as "apoptotic signaling pathways," "anatomical

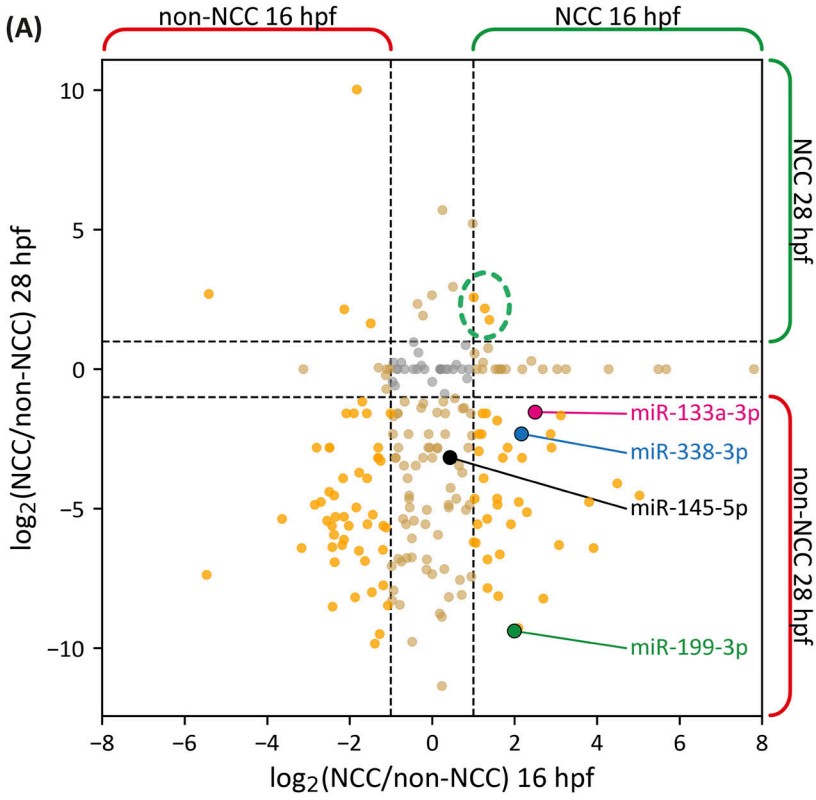

**Figure 1. Differential expression of neural crest miRNAs.**
**(A)** Identification of differentially expressed miRNAs in NCC and non-NCC from 16 hpf (x-axis) and 28 hpf (y-axis) transgenic zebrafish samples. Overrepresented miRNAs in either cell population are shown in yellow ($\log_2$ [NCC/non-NCC] > 2). miR-145-5p, miR-133a-3p, miR-199-3p, and miR-338-3p are highlighted in black, pink, green, and blue, respectively. For plotting, pseudocounts ($\epsilon$) were added to each normalized count value. See Table S1 and Fig S1 for non-pseudocount normalized values. The three miRNAs overrepresented in both time points are enclosed within a green dashed circle. **(B)** Quantification of miR-145-5p, miR-133a-3p, miR-199-3p, and miR-338-3p levels in NCC by RT−qPCR, compared with non-NCC; levels are presented as $\log_2$ (NCC/non-NCC). Statistical analysis was performed using a two-tailed $t$ test, *$P \leq$ 0.05, **$P \leq$ 0.01, n = 3, mean ± SEM. The circles correspond to the differential expression values obtained by the FACS/miRNA-seq experiment.

structure development," and "developmental processes," were also identified, though neuronal development terms were the most predominant. Similar results were obtained using Panther ($P <$ 0.05, FDR test), which also highlighted terms like "angiogenesis" and "axon guidance." DAVID ($P <$ 0.05, FDR test) revealed NC-related terms, such as "neural crest cell migration" and "cartilage development." No NC development process pathway was overrepresented in this gene list (KEGG). Using the bioinformatics tool STRING, which generates interaction networks from gene lists, we identified central node genes that participate in multiple processes or interact with several genes within the list. This approach allowed for a focused analysis of miR-133a-3p's impact on genes that may have a significant influence on zebrafish embryonic development. By analyzing a gene list with at least one putative miR-133a-3p binding site and expression data from ZFIN for NCC or their derivatives, other central node genes were elucidated, including *ddx21*, *pax9*, *sox10*, *erbb2*, and *gf20b* (Figs 2B and S2).

### miR-199-3p
For GO and KEGG pathway analyses, we extracted a list of 3,000 genes with predicted binding sites for miR-199-3p from TargetScan, with context+ scores between −0.04 and −9.1, although

a few overrepresented terms were found (Fig S3A, Tables S5 and S6). According to g:Profiler, 7 of 19 terms were related to nervous system development, whereas others were more general terms such as "multicellular organism development" and "developmental processes." Comparable results were obtained with Panther and DAVID, with terms like "axonogenesis" and "tissue development." A single KEGG pathway, "ErbB signaling pathway," was overrepresented. The absence of terms related to the NC or its derivatives does not necessarily indicate that miR-199-3p is uninvolved in NC development, as it may regulate only a few key genes essential for this process. STRING analysis, using genes with putative binding sites for miR-199-3p and expression reported in ZFIN for NCC or their derivatives, identified several central node genes, including *foxd3*, *wnt5b*, *erbb2*, and various genes in the FGF signaling pathway (Fig S3B).

### miR-338-3p
For GO analysis, we generated a list of 3,001 genes with predicted miR-338-3p binding sites with TargetScan, yielding context+ scores between −0.04 and −1.48 (Table S7). g:Profiler identified only five biological process terms, three related to neuronal development and two to positive regulation of biological processes (Fig S4A,

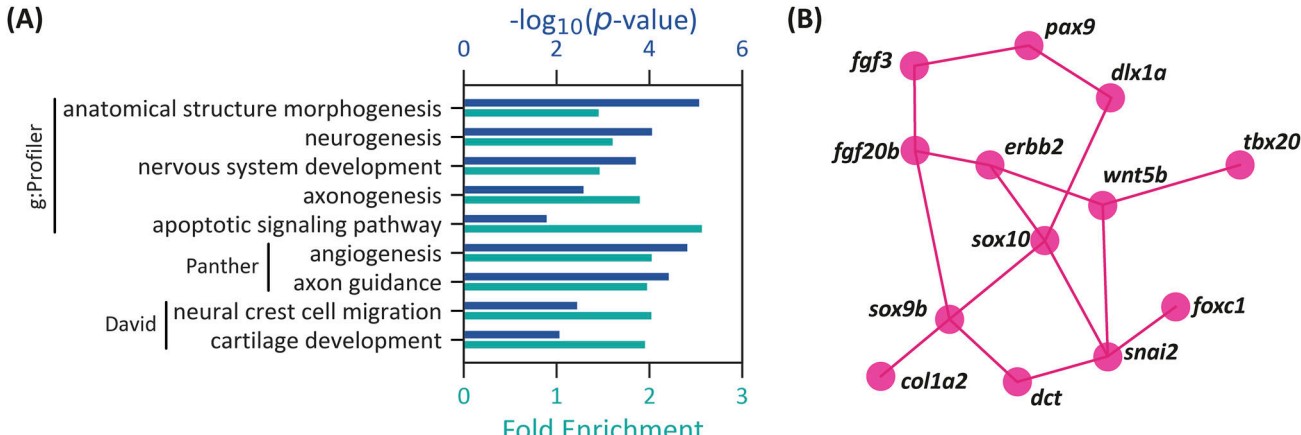

**Figure 2. GO and STRING analyses of miR-133a-3p putative target genes.**
Similar analyses for miR-199-3p and miR-338-3p are presented in Figs S3 and S4. **(A)** Selected overrepresented "Biological Processes" terms associated with miR-133a-3p putative target genes, as identified by g:Profiler, Panther, and DAVID. **(B)** Selected miR-133a-3p putative target genes and their predicted interactions based on STRING. For the complete interaction network, refer to Fig S2.

Table S8). Panther identified additional terms like "embryo development," whereas DAVID revealed terms such as "cartilage development," "fin morphogenesis," and "neural crest differentiation." No KEGG pathways were overrepresented. STRING analysis of genes with miR-338-3p binding sites and expression in NCC or derivatives identified *snai2*, *sox9b*, *kita*, and *pax2a* as central node genes (Fig S4B).

### miR-133a-3p, miR-199-3p, and miR-338-3p are differentially expressed during zebrafish embryogenesis

Previous zebrafish miRNA profiling studies have reported developmental regulation of miR-133, miR-199, and miR-338 family members (Chen et al, 2005; Kloosterman et al, 2006). However, quantitative, stage-resolved expression data for the specific mature miRNAs (miR-133a-3p, miR-338-3p, and miR-199-3p), and for across the early stages examined here remain limited and vary across datasets. We therefore analyzed their temporal expression from 5 hpf onward using RT–qPCR on whole-embryo extracts. Both miR-133a-3p and miR-199-3p showed the lowest expression at 5 hpf among the analyzed stages (Fig 3A and B). Their expression began to rise at 10 hpf, with a marked increase after 24 hpf. It should be noted that these profiles reflect global expression levels rather than NC-specific enrichment. Similarly, miR-338-3p was detected at 5 hpf (right after zygotic genome activation [Vastenhouw et al, 2019]), but its levels decreased between 10 and 24 hpf before increasing again (Fig 3C).

To prioritize putative targets, we filtered genes present in at least two or all three predicted miRNA putative target gene lists. Among these, only six NC-related genes were shared in all three lists, including *pdgfbb*, *tbx20*, and *ddx21* (see Table S9 for NC-related genes, and Table S10 for intersections). In addition, between 16 and 20 genes were common to two lists (Table S10). Several of the predicted target genes have been implicated in chondrocyte development, craniofacial structure formation, and melanophore differentiation.

To investigate the effects of miRNA gain of function, zebrafish embryos were microinjected with pri-miRNA constructs containing a tandem *dsRED* reporter sequence (dsRED-miR-133a, dsRED-miR-199, and dsRED-miR-338, Fig S5A). Control embryos received an mRNA encoding only the *dsRED* sequence (control-mRNA). We assessed during embryonic development both the presence of the reporter protein by fluorescence analysis and the pri-miRNA processing by stem–loop RT–qPCR. As expected, embryos collected at 24 hpf exhibited significant increases in miR-133a-3p, miR-199-3p, and miR-338-3p (hereafter called miR-133a, miR-199, and miR-338, respectively; Fig S5B–D). To ensure that the observed phenotypes were not secondary to general developmental delay or toxicity, we assessed gross morphology at 16 hpf and 48 hpf (Fig S5E–Q). Quantification of somite number (16 hpf) and eye size (16 and 48 hpf) revealed no significant global developmental defects across conditions. These results confirm the efficacy of our overexpression strategy and establish a robust foundation for assessing the impact of these miRNAs on neural crest derivative development.

### miR-133a and miR-338 overexpression alters melanophore development

Zebrafish pigmentation is determined by three types of NC-derived chromatophores: melanophores (black pigment), iridophores (iridescent pigment that reflects light), and xanthophores (yellow pigment) (Kelsh et al, 1996). In this study, we specifically examined the development of melanophores, which become visible around 25 hpf in the otic vesicle region. These cells subsequently expand across the dorsal head region (Fig 4A) and along the embryonic lateral line at 48 hpf (Fig 4E) (Kelsh et al, 1996). To investigate the role of miR-133a and miR-338, both having putative binding sites to genes involved in melanophore development (*dct*, *kita*, *tyrp1a*) (Greenhill et al, 2011), we performed a quantitative analysis on melanophore number present in 48-hpf larvae after the overexpression of each miRNA. The overexpression of either miRNA led

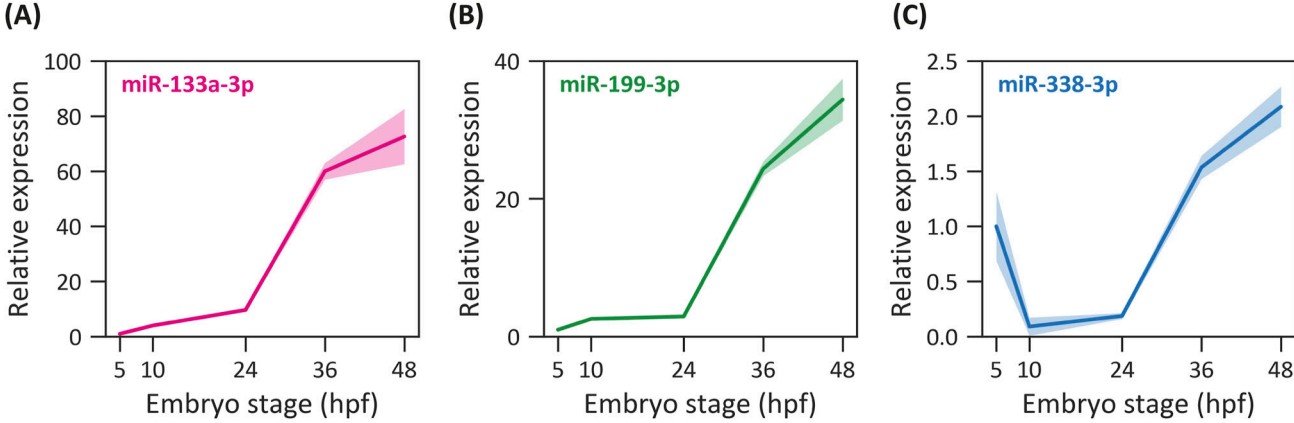

**Figure 3. miRNA expression profiles during zebrafish embryonic development.**
**(A, B, C)** RT–qPCR quantification of miR-133a-3p (A), miR-199-3p (B), and miR-338-3p (C) levels from 5 to 48 hpf. Normalized to 5-hpf stage, n = 3, mean ± SEM.

to a reduction in the melanophore number located in the dorsal head region (Fig 4B) and the lateral line (Fig 4F). Given that these miRNAs share putative target genes involved in melanophore development (like *dct*), we evaluated their potential synergistic effect through simultaneous overexpression. The co-overexpression of miR-133a and miR-338 produced a similar phenotype.

To explore the molecular basis of this phenotype, we analyzed the expression of *mitfa*, *kita*, *tyrp1a*, and *dct* (Greenhill et al, 2011). Notably, *mitfa* lacks predicted binding sites for either miRNA, and its expression remained unchanged after miR-133a or miR-338 overexpression (Figs 4C and S6D). In contrast, *dct* contains two predicted binding sites for miR-338 and multiple predicted sites for miR-133a (Fig S6E). Interestingly, although *dct* mRNA levels were unaffected in embryos overexpressing miR-338, they were significantly reduced upon miR-133a overexpression (Fig 4D). Both *kita* and *tyrp1a* contain one putative site for miR-338 (Fig S6F and G), and their respective mRNA levels were reduced in miR-338–overexpressing embryos, but not in miR-133a–overexpressing embryos (Fig 4G and H). For *kita*, this reduction was sustained upon combined overexpression. Altogether, these results show a complex and dynamic regulation, highlighting the redundancy typical of miRNA networks where multiple targets are shared.

## miR-133a and miR-338 overexpression adversely affects craniofacial cartilage development

To evaluate cartilage formation, embryos microinjected with dsRED-miRNA constructs were developed until 5 days post-fertilization (dpf), then fixed, and stained with Alcian Blue (Fig 5A and B). The overexpression of miR-199 did not produce any detectable changes in cartilage morphogenesis within the analyzed parameters (Fig 5D, H, and I). In contrast, miR-133a overexpression led to a significant reduction in Meckel's cartilage length and angle (ML and MAn, respectively; Fig 5C, F, and G), as well as a decreased ceratohyal angle (ChAn) and an increased ceratohyal-to-fin distance (ChD) (Fig 5F and G). Similarly, larvae overexpressing miR-338 exhibited shorter palatoquadrate

cartilages (PQ; Fig 5E and J), reduced Meckel's cartilage angles, and an increased ceratohyal angle (ChAn; Fig 5K).

Because miR-133a and miR-338 share several putative target genes associated with craniofacial development, we evaluated their potential synergistic effect. Our results revealed a significant reduction in both the Meckel's and ceratohyal angles, consistent with the trends observed when each miRNA was overexpressed individually (Fig 6A and B). In addition, co-microinjected larvae exhibited an increase in Meckel's cartilage area (Fig 6A and B). Interestingly, although the individual microinjection of dsRED-miR-133a or miR-338 led to reduced lengths of Meckel's (ML) and palatoquadrate (PQ) cartilages, and the increased distance from the fins to the ceratohyal cartilages (ChD), their dual overexpression restored these parameters to control levels.

To further elucidate the molecular mechanisms underlying the observed craniofacial defects, we quantified the mRNA expression levels of *runx3*, *sox10*, and *sox9b* (key regulators of cartilage development [Dalcq et al, 2012]) using RT–qPCR. This analysis revealed a significant reduction in *runx3* expression in embryos microinjected with dsRED-miR-133a and dsRED-miR-338, despite *runx3* containing only a single putative binding site for miR-133a (Figs 6C and S6C). Regarding *sox10*, mRNA levels were specifically reduced in dsRED-miR-133a–microinjected embryos, consistent with the presence of a predicted binding site for this miRNA in the 3'UTR of *sox10* (Figs 6D and S6A). In addition, the overexpression of both miR-133a and miR-338 resulted in decreased *sox9b* mRNA levels, likely because of the presence of multiple binding sites for these miRNAs in its 3'UTR (Figs 6E and S6B).

## miR-133a and miR-338 bind to the *sox9b* 3'UTR

To determine whether these mRNA reductions stem from direct regulation by miR-133a or miR-338, we performed a reporter assay using the *d4GFPn* gene (nuclear-localized destabilized eGFP, with a half-life of 4 h [Sánchez-Vásquez et al, 2019]) to assess *sox9b* posttranscriptional repression. *Sox9b* was selected because of its

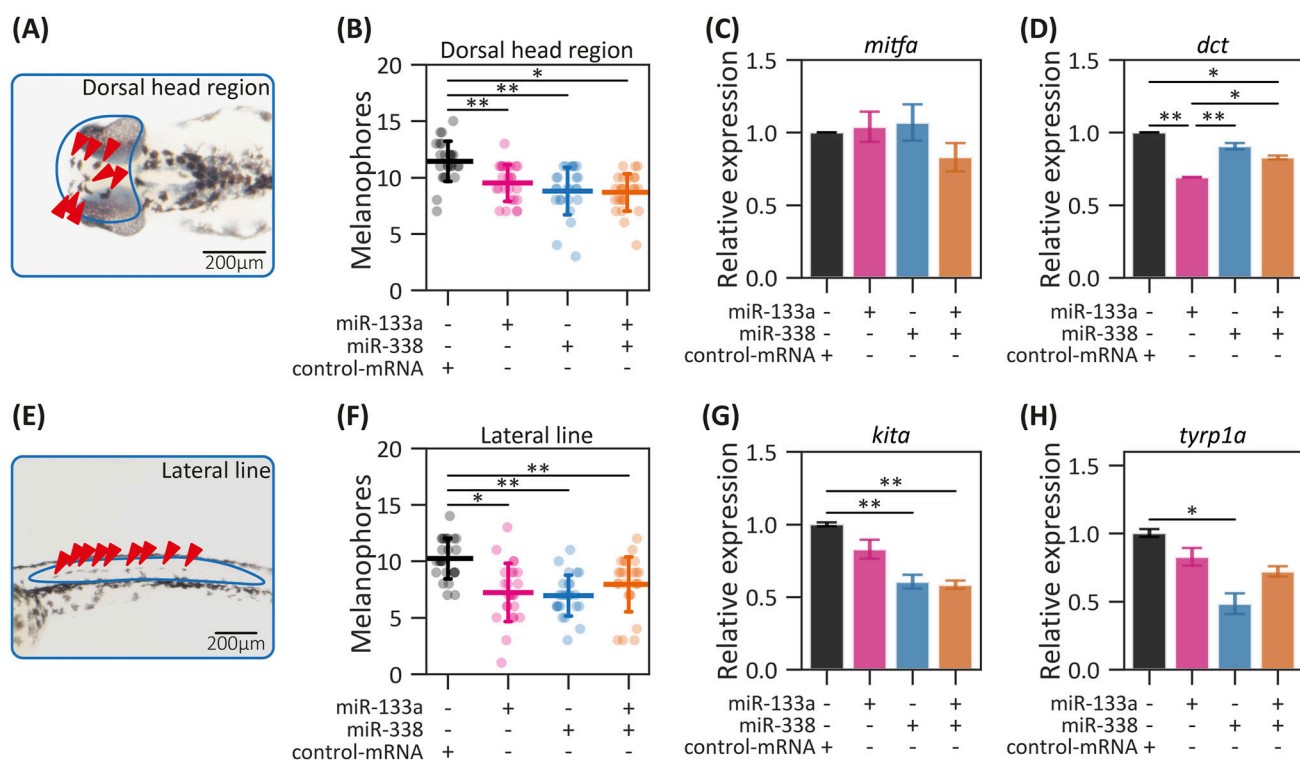

**Figure 4. Melanophore development in miR-133a– and miR-338–overexpressing embryos.**
**(A, B)** Representative images of melanophores (A) and quantification of melanophore numbers (B) in the head of 48-hpf larvae microinjected with dsRED-miR-133a, dsRED-miR-338, both, or control-mRNA. The analyzed areas are highlighted in blue, with counted cells indicated by red arrowheads. Scale bar (A): 200 µm. **(C, D)** mRNA expression levels of *mitfa* (C) and *dct* (D) in 24-hpf embryos microinjected with dsRED-miR-133a, dsRED-miR-338, both, or control-mRNA. **(E, F)** Representative images of melanophores (E), and quantification of melanophore numbers (F) in the lateral line of 48-hpf larvae microinjected with dsRED-miR-133a, dsRED-miR-338, both, or control-mRNA. The analyzed areas are highlighted in blue, with counted cells indicated by red arrowheads. Scale bar (E): 200 µm. **(G, H)** mRNA expression levels of *kita* (G) and *tyrp1a* (H) in 24-hpf embryos microinjected as described above. Statistical analysis (A, B, E, F): ANOVA followed by Tukey's post hoc test; *$P \leq 0.01$, **$P \leq 0.001$, n = 25, mean ± SD. Statistical analysis (C, D, G, H): ANOVA followed by Tukey's post hoc test; *$P \leq 0.05$, **$P \leq 0.01$, n = 3, mean ± SEM.

critical role in cartilage development and the presence of multiple predicted miRNA binding sites in its 3′UTR. Briefly, we used two sensor constructs: *sox9b*-3′UTR-1, which contains a single putative binding site for miR-133a and another for miR-338, and *sox9b*-3′UTR-2, which harbors two putative binding sites exclusively for miR-338 (Figs 7A and S6B). As a control, a third construct (*Xenopus* β-globin, control-3′UTR) was fused to the SV40-poly(A) sequence. The corresponding mRNAs were co-injected into one-cell-stage embryos alongside dsRED-miR-133a, dsRED-miR-338, both, or control-mRNA, and GFP fluorescence was quantified at the 50% epiboly stage (Fig S6H).

In embryos microinjected with *sox9b*-3′UTR-1, the overexpression of either miR-133a or miR-338 significantly suppressed *d4GFPn* fluorescence, with the strongest effect observed upon co-microinjection of both miRNAs (Figs 7C and S6J). Conversely, embryos injected with *sox9b*-3′UTR-2 exhibited a significant reduction in fluorescence only when dsRED-miR-338 was present (Figs 7D and S6K). No significant changes were detected in embryos microinjected with the control-3′UTR, regardless of miRNA treatment (Figs 7B and S6I). These findings align with RT−qPCR results, reinforcing the notion that miR-133a and miR-338 directly regulate *sox9b* expression by targeting its 3′UTR.

**miR-338 overexpression results in higher NCC numbers**

To quantitatively evaluate changes in the NCC population, we microinjected Tg(*sox10*:GFP) embryos with dsRED-miR-133a, dsRED-miR-338, or a control-mRNA, followed by flow cytometry analysis to quantify GFP-positive cells (Fig 8A). Although miR-133a overexpression did not significantly alter NCC numbers, miR-338 overexpression resulted in a notable increase in NCC abundance (Fig 8B). Interestingly, the co-expression of both miRNAs restored NCC numbers to control levels, suggesting a potential compensatory mechanism.

## Discussion

miRNAs are pivotal regulators of embryonic development and numerous diseases, including cancer, where they fine-tune gene expression at multiple levels (Kloosterman & Plasterk, 2006; Weiner, 2018). Because of their ability to target multiple mRNAs simultaneously (Kim, 2005) and considering the extensive network of transcription factors, enzymes, and structural proteins involved in the GRN (Gilbert, 2000; Martik & Bronner, 2017) of embryonic

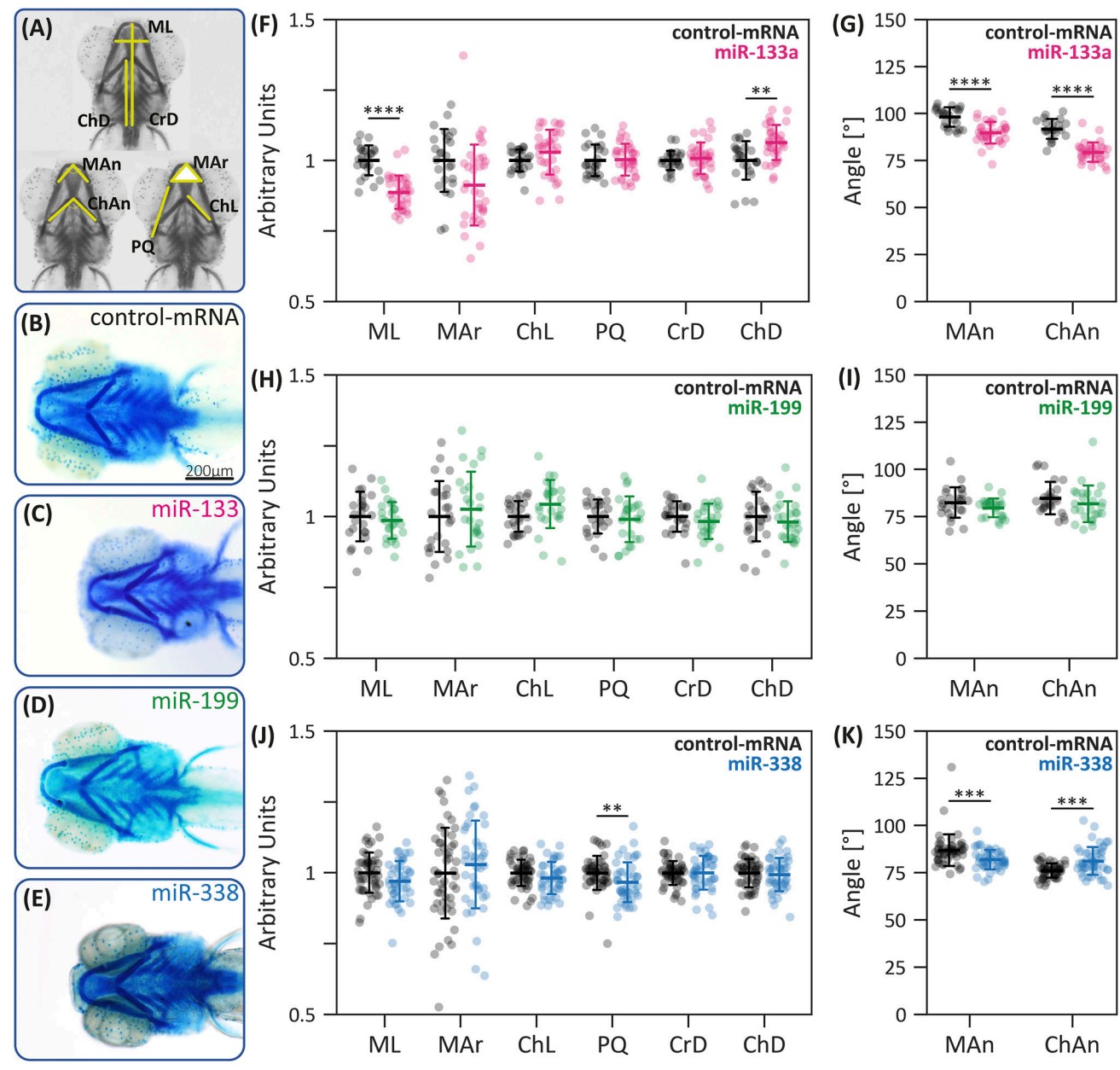

**Figure 5. Craniofacial development of zebrafish specimens overexpressing miR-133a, miR-199, and miR-338.**
**(A)** Schematic representation of the craniofacial parameters analyzed: ML, Meckel's cartilage length; MAr, Meckel's cartilage area; ChL, ceratohyal mean length; PQ, palatoquadrate mean length; CrD, cranial distance (from fins to Meckel's cartilage midpoint); ChD, ceratohyal distance (from fins to ceratohyal cartilage joint); MAn, Meckel's cartilage angle; ChAn, ceratohyal angle. **(B, C, D, E)** Representative ventral view images of Alcian Blue–stained cartilage in 5-dpf larvae, with the head positioned to the left. **(B, C, D, E)** Images show specimens developed from embryos microinjected with control-mRNA (B), dsRED-miR-133a (C), dsRED-miR-199 (D), and dsRED-miR-338 (E). **(F, G, H, I, J, K)** Craniofacial measurements of larvae microinjected with: dsRED-miR-133a (F, G), dsRED-miR-199 (H, I), and dsRED-miR-338 (J, K). Statistical analysis: two-tailed $t$ test; $*P \leq 0.05$, $**P \leq 0.01$, $***P \leq 0.001$, $****P \leq 0.0001$, n = 25 for miR-199, n = 22 for miR-133a, n = 48 for miR-338, mean ± SD. Scale bar in (B): 200 $\mu m$.

development, it is likely that many miRNA-mRNA interactions remain undiscovered. The parallels between NC development and cancer biology, particularly in processes such as EMT, migration, and invasion, suggest a shared regulatory landscape. Insights gained from one context can therefore enhance our understanding of the other.

Building on this concept, we selected miRNAs with established roles in tumorigenesis to investigate their regulatory influence on NC-derived tissues. Specifically, we focused on the roles of miR-133a-3p, miR-199-3p, and miR-338-3p, which are highly expressed in the premigratory NCC of *Danio rerio*. These miRNAs, which are conserved exclusively among vertebrates, have been implicated in cancer-related processes such as EMT, cytoskeletal reorganization, proliferation, migration, and apoptosis, as supported by both literature and bioinformatics analyses. Through this study, their contributions to NC-derived structure pathways were explored.

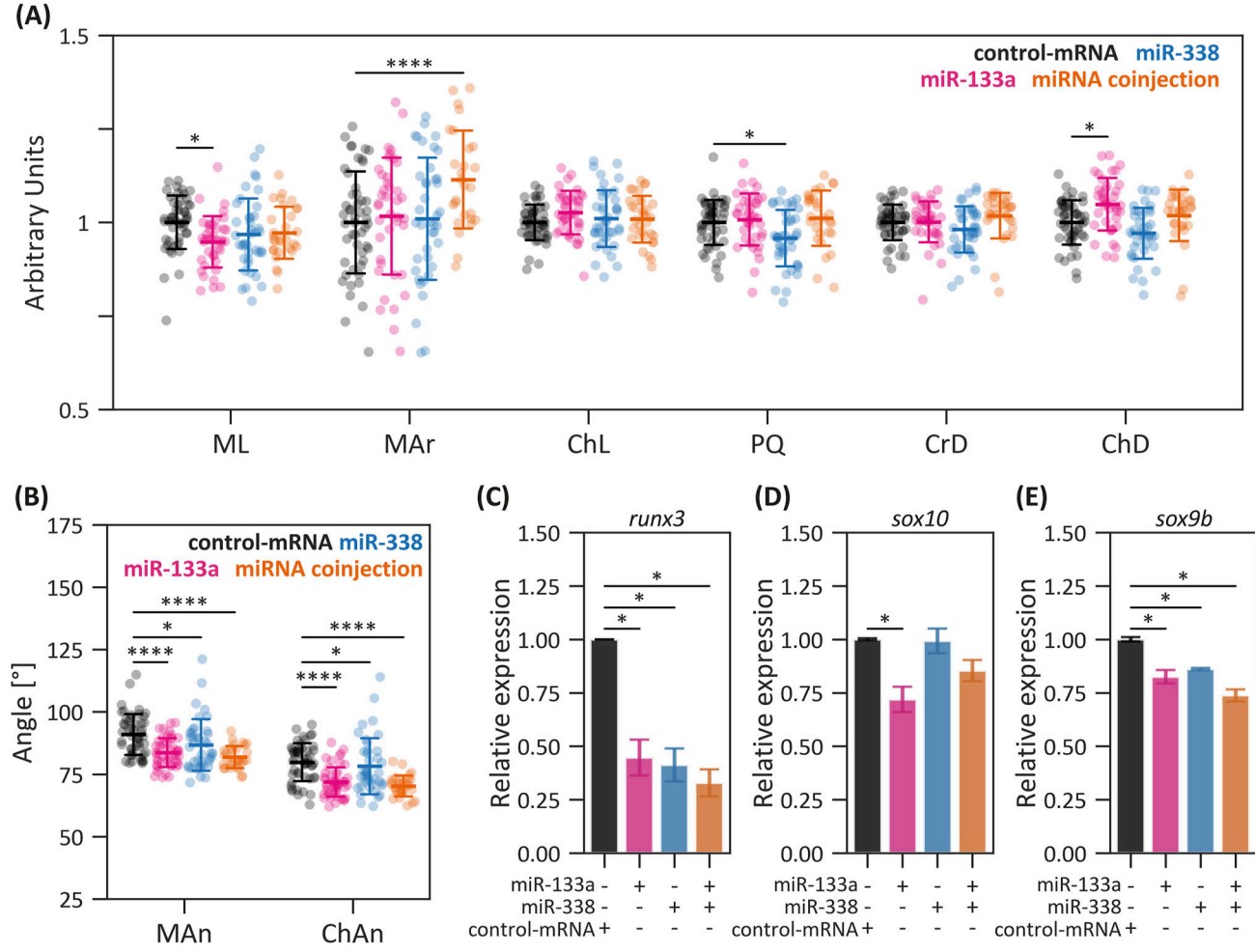

**Figure 6. Combined effect of miR-133a and miR-338 on craniofacial development.**
**(A, B)** Craniofacial measurements in larvae microinjected with dsRED-miR-133a, dsRED-miR-338, both, or control-mRNA. Statistical analysis: ANOVA followed by Tukey's post hoc test; *P ≤ 0.05, ****P ≤ 0.0001, n = 30, mean ± SD. **(C, D, E)** mRNA expression levels of *runx3* (C), *sox10* (D), and *sox9b* (E) in 24-hpf embryos microinjected with dsRED-miR-133a, dsRED-miR-338, both, or control-mRNA. Statistical analysis: ANOVA followed by Tukey's post hoc test; *P ≤ 0.05, n = 2, mean ± SEM.

To validate the developmental relevance of these selected miRNAs, we examined their temporal profiles. Although previous zebrafish profiling reported general developmental regulation of miR-133-3p and miR-199-3p families (Chen et al, 2005; Kloosterman et al, 2006), stage-resolved data for the specific isoforms examined here have been limited. Our analysis of whole-embryo lysates revealed that the up-regulation of miR-133a-3p and miR-199-3p begins at 10 hpf, around gastrulation. Published zebrafish expression datasets report dynamic regulation of NCC regulators such as *sox9b* and *foxd3* across these stages (White et al, 2017; Moreno et al, 2022), providing context for potential temporal overlap between these miRNAs and predicted targets. Because these measurements were performed on whole embryos, the later increase likely reflects cumulative expression from multiple tissues (including muscle), whereas the early rise is at least compatible with roles during early NCC programs. Similarly, miR-338-3p was detected as early as 5 hpf (immediately after zygotic genome activation [Vastenhouw et al, 2019]) and displayed a biphasic pattern during early development. In published datasets,

transcripts such as *sox9b* and *snai1b* are dynamically regulated across these stages (White et al, 2017; Moreno et al, 2022), which is consistent with the possibility of temporally coordinated miRNA-mRNA regulation during early embryogenesis. Overall, these observations support a potential role of miR-133a-3p, miR-199-3p, and miR-338-3p in modulating gene expression programs during early zebrafish development.

The literature suggests that miR-199 plays a role in regulating EMT and tumor cell migration, processes that are also crucial during early NC development (Giovannini et al, 2018; Wang et al, 2018; Zhang et al, 2019). However, no direct evidence has yet linked miR-199 to the regulation of specific NC derivatives. Notably, putative target genes of miR-199-3p include key NC specifiers such as *foxd3* (Martik & Bronner, 2017), as well as transcription factors involved in signaling pathways essential for NC specification, including WNT and FGFs (Schmidt et al, 2013). The lack of phenotypic defects after miR-199-3p overexpression serves as an important internal control, demonstrating that the injection conditions did not induce nonspecific toxicity or saturation of the RNAi machinery

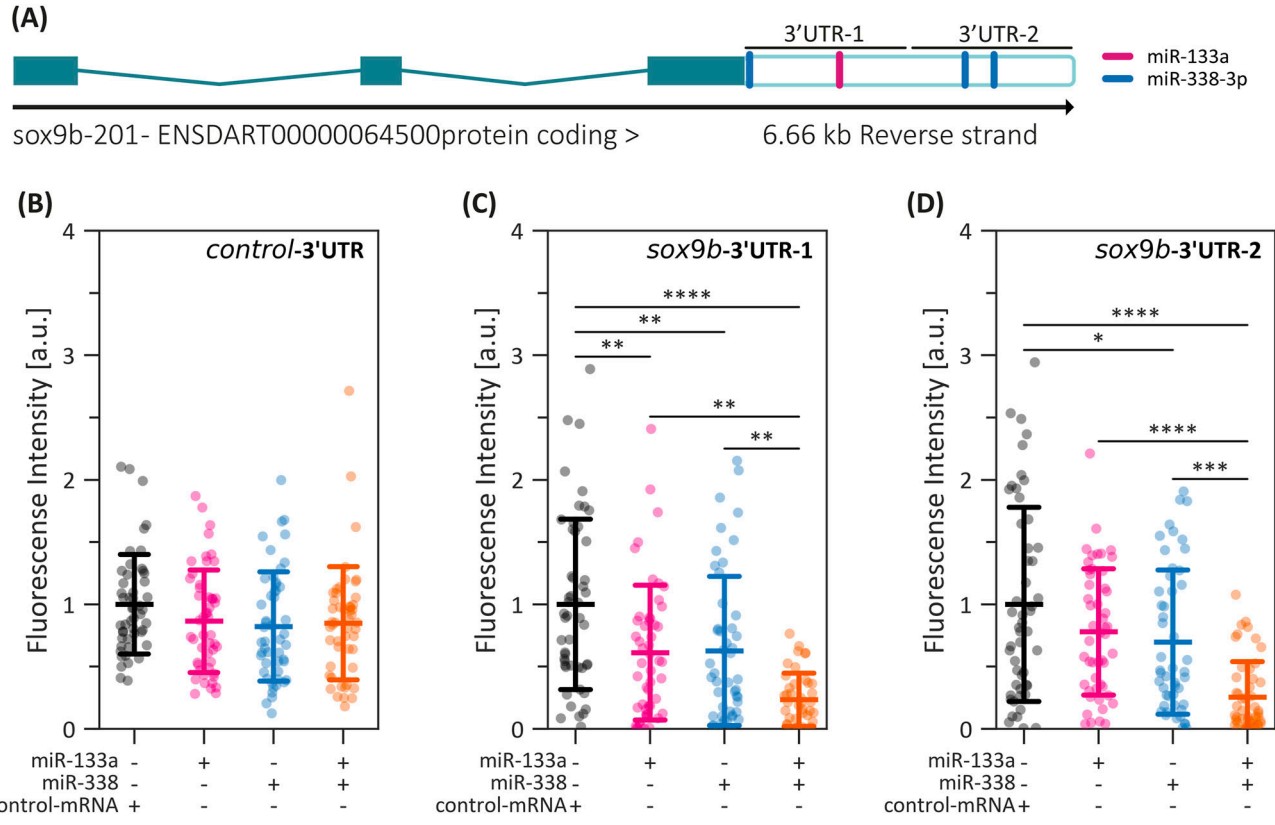

**Figure 7. Interaction between miR-133a, miR-338, and the sox9b 3′UTR.**
**(A)** Schematic representation of the sox9b gene structure (green), highlighting the 3′UTR (empty box) and the predicted binding sites for miR-133a (blue) and miR-338 (pink). The sox9b-3′UTR-1 and sox9b-3′UTR-2 fragments, used in the d4GFPn reporter constructs, are indicated. **(B, C, D)** Quantification of d4GFPn fluorescence in embryos co-injected with the control-3′UTR (B), sox9b-3′UTR-1 (C), or sox9b-3′UTR-2 (D), alongside dsRED-miR-133a, dsRED-miR-338, both, or control-mRNA. Statistical analysis was performed using ANOVA followed by Tukey's post hoc test; *P ≤ 0.05, **P ≤ 0.01, ***P ≤ 0.001, ****P ≤ 0.0001, n = 45, mean ± SD.

(Lund et al, 2011). Although miR-199-3p overexpression did not result in detectable craniofacial defects, its role in NC development cannot be ruled out, particularly in processes such as cell migration and iridophore differentiation. This possibility is supported by the presence of putative binding sites in genes implicated in these processes, such as *ltk* (Petratou et al, 2018), as well as its reported involvement in tumorigenesis and cell adhesion. Further studies, including functional assays and lineage tracing, are necessary to elucidate the precise role of miR-199-3p in NC development and its potential contribution to these biological processes.

The proper formation of craniofacial structures relies on the coordinated proliferation and positioning of cartilage progenitor cells (Yan et al, 2005). Transcription factors governing craniofacial development have been extensively characterized in *D. rerio*, mice, and cultured cells. Our findings indicate that miR-133a-3p and miR-338-3p regulate the expression of *sox9b*, a critical transcription factor required not only for craniofacial cartilage formation but also for melanophore differentiation. Here, we have demonstrated that these miRNAs repress *sox9b* expression at the mRNA level (consistent with mRNA destabilization) and at the reporter protein level (consistent with translational repression and/or mRNA decay) by directly interacting with its 3′UTR. The observed reduction

in *sox10*, *sox9b*, *dct*, and *runx3* mRNA levels could theoretically be attributed to indirect mechanisms, such as a decrease in the specific cell populations expressing these genes. However, our reporter assay results confirm that in the case of *sox9b*, miR-133a-3p and miR-338-3p exert their effects through a direct interaction with its 3′UTR. Because the reporter assay assesses protein levels (fluorescence) while our RT-qPCR data showed reduced mRNA, this suggests the regulation involves both mRNA degradation and translational repression, consistent with established miRNA mechanisms (Kloosterman & Plasterk, 2006). Notably, the co-overexpression of miR-133a-3p and miR-338-3p exacerbates craniofacial malformations, suggesting a synergistic effect on cartilage development. However, we also observed that the lengths of Meckel's and palatoquadrate cartilages (reduced in single injections) were restored to control levels upon co-overexpression. This suggests that the simultaneous presence of miR-133a-3p and miR-338-3p may exert a complex regulatory influence, potentially by regulating distinct sets of genes with opposing effects on this parameter, ultimately balancing one another's impact.

We also found that miR-133a-3p and miR-338-3p overexpression reduces expression levels of *runx3*, a crucial gene expressed in the pharyngeal endoderm that is essential for the activation of downstream cartilage GRN. The reduction in *runx3* expression after

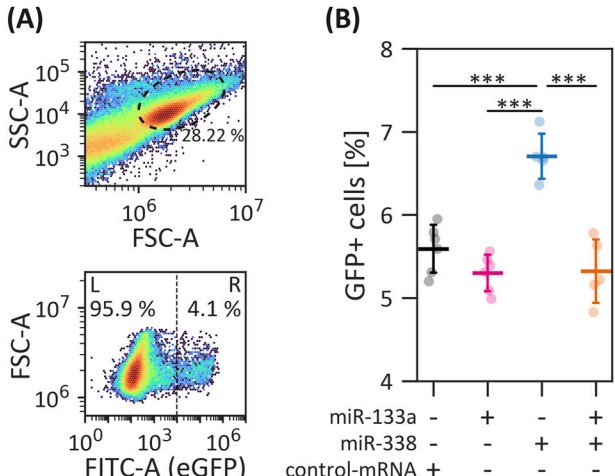

**Figure 8. Flow cytometry–based quantification of NCC.**
**(A)** Gating strategy used to identify zebrafish cells (upper panel) and discrimination of GFP-negative and GFP-positive cell populations (lower panel). **(B)** Proportion of NCC in 18-hpf embryos microinjected with dsRED-miR-133a, dsRED-miR-338, both, or control-mRNA. Statistical analysis: ANOVA followed by Tukey's post hoc test; ***$P \leq 0.001$, n = 5, mean ± SD.

miR-133a-3p and miR-338-3p overexpression mimics phenotypes observed in *runx3* morphants, including defects in viscerocranial and anterior neurocranial cartilage (Dalcq et al, 2012). Although miR-133a-3p likely directly targets *runx3*, miR-338-3p appears to act indirectly, potentially through the modulation of other transcription factors involved in chondrogenesis. These findings highlight the complexity of miRNA-mediated regulation, which can influence developmental pathways both directly and indirectly through highly interconnected GRNs. Furthermore, the specific reduction of *sox10* in miR-133a-3p–injected embryos likely contributes to these structural defects, as *sox10* is an early marker required for the specification of the NC populations that populate the pharyngeal arches (Lewis et al, 2004).

In melanogenesis, both miR-133a-3p and miR-338-3p also play significant regulatory roles. Vertebrate melanocytes (melanophores in fish) are critical for pigmentation, mating behaviors, individual recognition, and UV protection. They are also implicated in various pigmentation disorders, including albinism, vitiligo, and melanoma (Nordlund et al, 2007). Our data indicate that miR-133a-3p overexpression reduces *dct* expression, which may account for the observed decrease in melanophore numbers. Interestingly, although *sox10* is required for the activation of *mitfa* (Greenhill et al, 2011), we did not detect a corresponding reduction in *mitfa* mRNA levels at 24 hpf. Because *mitfa* lacks predicted binding sites for miR-133a-3p and miR-338-3p, the absence of a *mitfa* decrease likely reflects indirect buffering/compensatory regulation within the melanophore gene regulatory network and/or limited sensitivity of whole-embryo measurements at 24 hpf, given the restricted expression domain of *mitfa* relative to *sox10/sox9b*. Because *sox9b* is transiently required in melanophore precursors, the down-regulation of *sox9b* by miR-133a-3p and miR-338-3p may disrupt early melanophore activation, ultimately leading to reduced melanogenesis. The absence of detectable changes in the

expression levels of *dct* (an enzyme required for melanin biosynthesis (Greenhill et al, 2011)) after miR-338-3p overexpression does not necessarily exclude direct miRNA-mediated regulation. It is possible that miR-338-3p modulates *dct* expression at the translational level rather than by inducing mRNA degradation, a mechanism consistent with the well-established dual role of miRNAs in posttranscriptional regulation. Further studies are needed to elucidate the precise molecular pathways through which these miRNAs influence zebrafish melanophore differentiation and pigmentation.

Contrary to its well-documented tumor suppressor role (Huang et al, 2011, 2015; Liu et al, 2015), miR-338-3p overexpression led to an increased population of NC progenitor cells in zebrafish embryos. This expansion, coupled with the concomitant effect on melanophores and cartilage structures, suggests a significant block in lineage commitment. Mechanistically, this effect likely stems from miR-338-3p's regulatory influence on genes involved in cell proliferation, such as the predicted targets *wif1* and *ddx21*, both of which have been implicated in NC expansion and developmental patterning (Yin et al, 2012a; Guglielmi et al, 2020), alongside the down-regulation of key pro-differentiation factors *sox9b* and *sox10*. By fine-tuning the *sox9b*/*sox10* regulatory axis, miR-338-3p likely maintains NCC in a proliferative, undifferentiated state, preventing the transition from multipotent progenitors to specialized derivatives (Stevanovic et al, 2022).

The dual role of miR-338-3p in promoting proliferation while inhibiting terminal differentiation echoes functions seen in tumorigenesis, where the subversion of differentiation programs is a hallmark of malignancy. Interestingly, the co-expression of miR-133a-3p restored NCC numbers to control levels, suggesting a compensatory mechanism between the two miRNAs. In contrast, because miR-133a-3p overexpression did not affect NCC proliferation, the associated defects in melanophore and craniofacial development are more likely because of defects in specification and differentiation rather than alterations in progenitor cell number. This aligns with previous reports establishing miR-133a-3p as a key regulator of cell fate decisions and differentiation rather than proliferation in other developmental contexts (Ivey & Srivastava, 2010; Garcia-Padilla et al, 2022; Sharma et al, 2023).

Although GO analyses did not reveal extensive enrichment of NC-related pathways, our findings strongly suggest that miR-133a-3p and miR-338-3p play crucial roles in NCC development. These miRNAs likely exert their effects through both direct and indirect regulatory mechanisms, influencing key transcription factors such as *sox9b* and *runx3*. The interplay between these miRNAs and their target genes underscores their broader significance in cellular plasticity and differentiation.

It is important to note that bioinformatics predictions of miRNA-mRNA interactions often lack experimental validation. Demonstrating the co-expression of miRNAs and their targets within the same cell type is essential for confirming functional interactions (Xu et al, 2013; Yu et al, 2016; Sánchez-Vásquez et al, 2019; Lukiw & Pogue, 2020). Future studies employing in situ hybridization will be valuable for validating the co-localization of miRNAs and their putative targets, such as *sox10*, *runx3*, and *dct*. The intricate overlap of GRNs regulating various NC derivatives suggests that miRNA overexpression likely affects multiple tissues simultaneously. For

instance, the dual regulatory effect of miR-133a-3p and miR-338-3p on *sox9b* influences both craniofacial cartilage formation and melanogenesis, demonstrating how miRNAs orchestrate broad developmental processes.

In conclusion, our findings highlight the essential role of precise miR-133a-3p and miR-338-3p regulation in the development of NC-derived structures in *D. rerio*, particularly craniofacial cartilages and melanophores. The craniofacial abnormalities observed in miRNA-overexpressing embryos likely stem from the direct repression of *sox9b* and *runx3* expression, whereas the reduction in the number of melanophores may result from down-regulation of *dct*. Beyond their developmental roles, the shared regulatory features between NC formation and tumorigenesis underscore a broader function for these miRNAs in modulating cellular plasticity, proliferation, and differentiation. These insights not only deepen our understanding of miRNA-mediated control during vertebrate embryogenesis but also highlight the potential of miR-133a-3p and miR-338-3p as therapeutic targets in neuro-cristopathies and cancer. Future studies should further investigate their mechanistic roles and translational relevance in both developmental and pathological contexts.

# Materials and Methods

### Zebrafish care

All zebrafish were handled in accordance with national and international guidelines (Westerfield, 2000). All protocols involving zebrafish from the Calcaterra Lab were approved in advance by the *Comité de Bioética para el Manejo y Uso de Animales de Laboratorio* of the Facultad de Ciencias Bioquímicas y Farmacéuticas, UNR (files 6,060/374, resolution 207/2018). Adult zebrafish were maintained at a constant temperature of 27°C ± 1°C under a 14:10-h light/dark cycle (Westerfield, 2000). WT fish from the AB strain (ZIRC, Oregon University) were used in this study. The transgenic lines used included Tg(*actb1*:eGFP), Tg(*sox10*:eGFP, *sox10*:mRFP) (Kwak et al, 2013), and Tg(*sox10*:eGFP).

Zebrafish embryos were obtained according to the methodology described by Kimmel et al (1995). Briefly, selected adults were kept overnight at 28.5°C in breeding tanks, with males and females separated in a 3-to-4 ratio, respectively. Embryos were staged by visualization under a stereoscopic microscope (Olympus MVX10 stereoscopic microscope and Olympus C-60 ZOOM digital camera) based on morphological development in hours or days post-fertilization (hpf or dpf) at 28°C (Kimmel et al, 1995).

### Flow cytometry and FACS

Embryos at 16 and 28 hpf were manually dechorionated and deyolked using Ginsburg buffer (111.22 mM NaCl, 3.35 mM KCl, 2.7 mM CaCl$_2$, 2.38 mM NaHCO$_3$) through gentle pipetting and vortexing (600 rpm, 5 min, MX-S Vortex Spin, DLab). The samples were then centrifuged at 2,500 rcf for 3 min at 4°C, resuspended in 1× PBS containing 0.125% (wt/vol) trypsin, and incubated at 37°C for 15 min to facilitate dissociation. After incubation, the samples were

centrifuged (500 rcf, 15 min, 4°C), resuspended in 500 µl 1× PBS, filtered through a 40-µm mesh, and counted.

To calibrate fluorescence-based cell sorting, WT and Tg(*actb1*:eGFP) lines were used as controls, representing 0% and 100% eGFP+ cells, respectively. For FACS and flow cytometry cell counting, the transgenic lines Tg(*sox10*:eGFP, *sox10*:mRFP) and Tg(*sox10*:eGFP) were employed, respectively. FACS was conducted using BD FACSAria II and FACSAria III systems (BD Biosciences) at a flow rate of 30 µl/min. GFP+ cells were used to establish gating parameters for zebrafish cells (Fig S1A) (Gallardo & Behra, 2013), whereas mRFP fluorescence served as a control, achieving >99% mRFP+ within the eGFP+ population. Cells sorted by FACS were directly collected in TRIzol (Invitrogen) for subsequent RNA extraction. Sorting efficiency was analyzed by measuring NCC marker *foxd3* and non-NCC marker *tbx5a* in each population (Fig S1D).

Flow cytometry cell counting was conducted using a BD Accuri C6 Plus system (BD Biosciences), with a flow rate of 30 µl/min, ensuring a minimum of 100,000 events were recorded per condition. Each condition was analyzed in triplicate for each biological replicate. Data processing and analysis were performed using FlowJo software (BD Biosciences).

### RNA-seq

Total RNA was extracted using TRIzol reagent (Life Technologies) with DNase treatment to remove genomic contamination. RNA concentration and purity were assessed using the Qubit RNA BR Assay kit (Thermo Fisher Scientific), whereas RNA integrity was evaluated with RNA 6000 Nano Bioanalyzer 2100 Assay (Agilent). After confirming RNA quality, samples from different experimental groups were used for library construction at each developmental stage.

Small RNA libraries were prepared using the TruSeq Small RNA Library Prep kit (Illumina) according to the manufacturer's protocol. Sequencing was performed on an Illumina NextSeq High Output platform in paired-end mode, generating 75-bp reads. Image analysis, base calling, and quality scoring were carried out using Illumina's Real-Time Analysis software (v3.4.4). The cleaned-up reads were checked using FASTQC and then aligned to the zebrafish genome using Bowtie (Langmead et al, 2009). Aligned reads were counted using HTSeq (Putri et al, 2022). Read counts for each sample included in the study were normalized, after which the miRNA differential expression was analyzed using DESeq2 (Love et al, 2014). For plotting purposes, pseudocounts ($\epsilon$) were added to each normalized count value. The pseudocount was calculated as half of the minimum nonzero normalized count across all samples, resulting in a value of $\epsilon = 0.0438$ (see Table S1 and Fig S1 for non-pseudocount normalized values).

### Bioinformatics

Zebrafish (*D. rerio*) pre-miRNA sequences were retrieved from Ensembl (Release: 98, Assembly: GRCz11) and miRBase (version: 22.1). Putative miRNA target genes were identified using TargetScan Fish (targetscan.org/fish_62) (Ulitsky et al, 2012). Gene expression data for zebrafish were extracted from the EMBL-EBI Expression

Atlas, based on experimental data from White et al (2017) and Moreno et al (2022).

Gene Ontology (GO) and KEGG metabolic pathway analyses were conducted using multiple bioinformatics tools: DAVID (version: v2022q4; GO and KEGG, FDR and $P < 0.05$) (Huang et al, 2009), PANTHER (version: The Gene Ontology Consortium, 2021; GO, FDR and $P < 0.05$) (Mi et al, 2019), and g:Profiler (version: e111_eg58_p18_f463989d; GO and KEGG, g:SCS threshold and $P_{adj} < 0.05$) (Raudvere et al, 2019).

Statistical analyses were performed using Prism 9.5 (GraphPad Software). Depending on the experimental design, either a two-tailed $t$ test or ordinary/nested ANOVA followed by Tukey's post hoc test was applied. Statistical significance was set at $P \leq 0.05$. Data visualization was performed with Prism 9.5 or with custom scripts using Python (v3.10), generated with pandas, seaborn, and matplotlib libraries.

### Reverse transcription followed by quantitative PCR (RT–qPCR)

For RNA extraction, 35–45 embryos per condition were collected at the required developmental stage and rapidly flash-frozen in liquid nitrogen. Samples were either processed immediately or stored at –80°C for long-term preservation. Embryos were homogenized in TRIzol, followed by chloroform extraction and isopropanol precipitation. The total RNA concentration was determined by measuring absorbance at 260 nm using a NanoVue spectrophotometer (GE Healthcare).

Reverse transcription was performed using M-MLV reverse transcriptase (Promega) with 1 µg of RNA. Oligo-dT primers were used for mRNA reverse transcription, whereas specific stem–loop (Kramer, 2011) oligonucleotides were designed for each miRNA. Quantitative PCR (qPCR) was conducted using HOT FIREPol EvaGreen qPCR Mix Plus (Solis BioDyne) on a RealPlex4 thermocycler (Eppendorf). Zebrafish *rpl13* and *eef1a1l1* cDNAs were amplified as internal references. Primer sequences are provided in Table S11. Data analysis was carried out using REST 2009 software (QIAGEN) (Pfaffl et al, 2002), following MIQE guidelines (Bustin et al, 2009) to ensure experimental accuracy and reproducibility.

### miRNA overexpression

For the overexpression experiments, the genomic regions encoding zebrafish *miR-133a-3p* (miRBase Accession: MIMAT0001830; chr2: 4,113,889–4,114,465), *miR-199-3p* (miRBase Accession: MIMAT0003155; chr5: 1,376,464–1,377,047), and *miR-338-3p* (miRBase Accession: MIMAT0048673; chr3: 51,907,303–51,907,638) were PCR-amplified and cloned into the pSP64T-dsRED expression vector using *Eco*RI and *Xho*I restriction sites (primer sequences provided in Table S11, with restriction sites in lowercase).

For mRNA synthesis, plasmids were linearized with either *Xba*I or *Bam*HI (Invitrogen) and transcribed using the mMESSAGE mMACHINE SP6 transcription kit (Invitrogen). A dsRED-control vector lacking miRNA sequences was prepared as a negative control.

Microinjections were performed at the one-cell stage, targeting the yolk just beneath the blastomere, using a gas-driven microinjection system (MPPI-2 Pressure Injector, Applied Scientific Instrumentation). Each embryo received 1.25 ng of transcript and was

maintained at 28°C until the required developmental stage for analysis.

### Phenotype observation

Zebrafish embryos (~16 hpf) and larvae (48 hpf) from each treatment group were collected and immobilized in 3% (wt/vol) methylcellulose containing 0.15 mg/ml tricaine (ethyl 3-aminobenzoate methanesulfonate salt, A5040; Sigma-Aldrich). Specimens were oriented laterally and imaged using an Olympus MVX10 stereomicroscope equipped with an Olympus C-60 ZOOM digital camera. The somite number was counted manually. Eye surface area was quantified by outlining the eye perimeter using the freehand selection tool in FIJI (National Institutes of Health, Bethesda, MD, USA). Measurements were normalized to the mean of the control group.

### Reporter assay

The *sox9b* d4EGFPn-3′UTR reporter constructs were originally generated by Steeman et al (2021). An empty pSP64T-d4EGFPn vector with *Xenopus* β-globin (*hbg1*, ENSXETG00000025667) 3′UTR was used as a control. Transcripts were microinjected into one-cell-stage embryos at a concentration of 1.5 ng, along with 1.25 ng of dsRED-miR-133a, dsRED-miR-338, or control-dsRED (control-mRNA) transcripts. This study was repeated twice; each trial was conducted separately and individually involved the microinjection of more than 45 embryos per condition.

Fluorescence from d4EGFPn was assessed at the 50% epiboly stage. At least 40 embryos per condition were imaged using an Olympus MVX10 stereomicroscope equipped with an Olympus C-60 ZOOM digital camera (Olympus). Fluorescence intensity was quantified using QuantiFish software (Stirling et al, 2020).

### Alcian Blue staining

To assess the effects on cranial structures, 30 larvae at 5 dpf were fixed overnight at 4°C in 4% (wt/vol) PFA prepared in 1× PBT (1× PBS with 0.1% [vol/vol] Tween-20). After fixation, samples were washed four times with 1× PBT and stained according to previously established protocols (Weiner et al, 2019).

Images were captured using an Olympus MVX10 stereomicroscope equipped with an Olympus C-60 ZOOM digital camera. Cranial cartilage parameters were measured using ImageJ software (National Institutes of Health, Bethesda, MD, USA) (Schneider et al, 2012) following previously reported methods (Weiner et al, 2020).

### Melanophore counting

Larvae at 48 and 72 hpf were anesthetized using 0.15 mg/ml tricaine and positioned in petri dishes containing 3% (wt/vol) methylcellulose. Using two 30 G needles, larvae were carefully oriented to obtain lateral and dorsal images under an Olympus MVX10 stereomicroscope equipped with an Olympus C-60 ZOOM digital camera. Imaging was performed using top illumination against a white background. Pigmented cell counts were conducted manually by analyzing the acquired images with Fiji software.

## Data Availability

The miRNA-seq data from this publication have been deposited in the NCBI Gene Expression Omnibus (GEO) database (https://www.ncbi.nlm.nih.gov/geo/) under the Accession number GSE300363.

## Supplementary Information

## Acknowledgements

We thank Sebastián Graziati for expert fish care, Silvana Sut for laboratory assistance, and Rodrigo Vena for microscopy assistance. TJ Steeman and M Torres are fellows of CONICET, JA Rubiolo and G Coux are staff member of CONICET, and AMJ Weiner and NB Calcaterra have been staff members of CONICET and Universidad Nacional de Rosario. AMJ Weiner is currently employed at BBD BioPhenix, Spain. This work was supported by Agencia Nacional de Promoción Científica y Tecnológica, grant number PICT 2016-0914 to AMJ Weiner, and Consejo Nacional de Investigaciones Científicas y Técnicas, grant numbers PIP 2015-2017-11220150100170CO to NB Calcaterra, and PIP 2021-2023-11220200100505CO to G Coux.

### Author Contributions

TJ Steeman: conceptualization, data curation, formal analysis, validation, investigation, visualization, methodology, and writing—original draft, review, and editing.

M Torres: data curation, investigation, methodology, and writing—review and editing.

AMJ Weiner: conceptualization, resources, data curation, funding acquisition, investigation, methodology, and writing—review and editing.

JA Rubiolo: data curation, formal analysis, investigation, and writing—review and editing.

LE Sánchez: resources, supervision, and writing—review and editing.

G Coux: resources, supervision, and writing—review and editing.

NB Calcaterra: conceptualization, supervision, funding acquisition, and writing—original draft, review, and editing.

### Conflict of Interest Statement

The authors declare that they have no conflict of interest.

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
