## [Reviewer comments · Life Science Alliance]

miR-133a-3p and miR-338-3p Shape Neural Crest Derivatives in Zebrafish

Tomás Steeman, Mercedes Torres, Andrea Weiner, Juan Rubiolo, Laura Sánchez, Gabriela Coux, and Nora Calcaterra
DOI: <https://doi.org/10.26508/lsa.202503431>

Corresponding author(s): Tomás Steeman, Instituto de Biología Molecular y Celular de Rosario

Review Timeline:

Submission Date:	2025-06-23
Editorial Decision:	2025-08-11
Revision Received:	2026-03-06
Editorial Decision:	2026-04-01
Revision Received:	2026-04-06
Accepted:	2026-04-06

Scientific Editor: Tim Fessenden

Transaction Report:

August 11, 2025

Re: Life Science Alliance manuscript #LSA-2025-03431

Dr. Tomás José Steeman
Instituto de Biología Molecular y Celular de Rosario
Ocampo y Esmeralda
Rosario 2000
Argentina

Dear Dr. Steeman,

Thank you for submitting your manuscript entitled "miR-133a and miR-338-3p Shape Neural Crest Derivatives in Zebrafish" to Life Science Alliance. The manuscript was assessed by expert reviewers, whose comments are appended to this letter. We sincerely appreciate your patience during the unusually long review process.

As you will see, both reviewers appreciated the new observations on gene expression regulation of the neural crest lineage by non-coding RNAs. However, Reviewer 1 remarked on important validation and controls needed to fully support claims. In particular, confirming successful FACS isolation of NCCs (point 1), and suitable controls for miRNA overexpression (points 3 and 4) and for injection assays (point 6). Reviewer 2 requested important validation by QPCR of key results (points 2 and 3). We invite you to submit a revised manuscript addressing the Reviewer comments, focusing on the above points.

Thank you for this interesting contribution to Life Science Alliance. We are looking forward to receiving your revised manuscript.

Sincerely,

-- Summary blurb (enter in submission system): A short text summarizing in a single sentence the study (max. 200 characters including spaces). This text is used in conjunction with the titles of papers, hence should be informative and complementary to the title and running title. It should describe the context and significance of the findings for a general readership; it should be

written in the present tense and refer to the work in the third person. Author names should not be mentioned.

B. MANUSCRIPT ORGANIZATION AND FORMATTING:

Reviewer #1 (Comments to the Authors (Required)):

In this manuscript, the authors investigate the role of microRNAs (miRNAs) in neural crest cells (NCCs) using zebrafish as a model organism. To this end, the authors isolated NCCs and non-NCCs by fluorescence-activated cell sorting (FACS), followed by RNA-Seq analysis at two developmental stages (16 hpf and 18 hpf). Based on miRNA enrichment in the NCC population and subsequent gene network analyses of putative target genes, the authors focused on miR-133a, miR-199-2, and miR-388 for functional studies. Overexpression of each miRNA, alone or in combination, resulted in phenotypes affecting pigment cells and craniofacial cartilage development. The authors also identified several mRNA targets of these miRNAs through computational prediction and reporter assays.

While the identification of these three miRNAs as potential regulators of NCC development is of interest, the manuscript is not well organized and is often difficult to follow. The data presented are generally of good quality, but several experiments lack appropriate controls and validation steps.

In this reviewer's opinion, the manuscript is currently premature and requires significant reorganization and clarification before it can be considered a solid research article.

Major points

1. The authors isolated NCCs and non-NCCs by FACS from Tg(sox10:eGFP,sox10:mRFP) embryos. However, validation of this approach, which is critical to the subsequent analysis, was not provided. First, a representative FACS plot should be shown. Second, the purity of the sorted NCC and non-NCC populations should be validated, for example, by RT-PCR of established marker genes for neural crest and other cell types.

Expected timeframe: one-two months.

2. The rationale for selecting the three miRNAs for functional analysis is not clearly or systematically presented. In the Results section, each miRNA is preceded by a lengthy introductory paragraph summarizing previous findings; however, much of this content is descriptive and not essential to justifying their selection. A more concise and focused explanation of why these particular miRNAs were chosen would improve clarity.

3. The functional analysis of the three miRNAs relies primarily on overexpression experiments, which are known to potentially induce artificial interactions between miRNAs and mRNAs containing seed-matched sites. To strengthen the conclusions drawn from these experiments, complementary loss-of-function approaches-such as genetic mutants or antisense oligonucleotides-are necessary to validate the physiological relevance of the observed effects.

Expected timeframe: three-six months.

4. Related to the above point, the choice of control in overexpression experiments is also critical. Using a control mRNA lacking a miRNA sequence is insufficient. A more appropriate control would be the overexpression of an unrelated miRNA or miRNA harboring mutations in the seed region, to account for potential non-specific effects. Overexpression can lead to saturation of Argonaute (Ago) proteins and induce off-target phenotypes, as previously reported (PMID: 21576259). Careful experimental design is required to distinguish specific miRNA function from such artifacts.

Expected timeframe: two-three months.

5. Regarding the phenotypic analysis, bright-field images of whole embryos at lower magnification are warranted to assess overall body morphology. Such views would help determine whether the observed phenotypes are specific or part of broader developmental defects, and would strengthen the interpretation of the results.

Expected timeframe: one month.

6. In Figure 7G, the reporter analysis lacks an internal control for quantification. Injection-based assays are inherently less precise than techniques such as micropipetting, making normalization essential. To ensure reliable quantification-especially for the subtle effects typically observed with miRNA-mediated regulation-it is necessary to include an internal control, such as a co-injected mRNA encoding an independent reporter protein.

Expected timeframe: one-two months.

7. To further support the direct interaction between miRNAs and their target mRNAs, reporter assays using constructs with mutated target sites would be desirable. Including such experiments would help demonstrate target site specificity and strengthen the overall interpretation.

Expected timeframe: one-two months.

8. The figures lack figure number labeling, which significantly disrupts the flow of reading and makes the manuscript difficult to follow.

Minor points

1. On page 2, line 72, "The miRISC complex targets mRNA by binding to a specific "seed region" of the miRNAs" This is incorrect. The miRISC complex targets mRNAs through the binding of a specific "seed region" of the miRNA to complementary sites within the mRNA.

2. On Page 5, line 200, the statement "Since the expression profiles of miR-133a, miR-199-2, and miR-338-3p in zebrafish remain incompletely characterized" is inaccurate. Even early studies on miRNA profiling in zebrafish, such as the one by Wienholds et al. (PMID: 15937218), reported temporal expression data for miR-133 and miR-199. Furthermore, numerous follow-up studies have since provided detailed expression profiles of these miRNAs. The authors should revise this statement to accurately reflect the existing literature.

3. In several instances, the authors appear to conflate miRNA targets with target candidates or putative targets based solely on prediction. For example, on page 6, line 216, the phrase "Several of the identified target genes have been..." is misleading, as the discussion refers to predicted candidates rather than experimentally validated targets. It is important to clearly distinguish between computational predictions and experimentally confirmed miRNA-mRNA interactions, as the latter requires direct validation.

In addition, the manuscript inconsistently refers to translational repression in contexts where the observed effects may reflect a combination of translational repression and mRNA decay. While the authors briefly acknowledge this distinction, their terminology is not used consistently throughout the manuscript. To avoid overinterpretation, the authors should revise these statements for clarity and accuracy.

4. On page 6, line 245" This aligns with the role of miRNAs as fine tuners of gene expression, where their regulatory impact does not necessarily scale linearly with increased expression." Citation is required for this specific statement.

5. On page 8, line 307, the term "d4GFPn" is introduced without explanation. All specific terms, constructs, or materials should be clearly defined at their first appearance to ensure clarity for readers who may not be familiar with them.

Reviewer #2 (Comments to the Authors (Required)):

The work by Steeman et al. investigates the role of microRNAs as modulators of neural crest cell lineages. Using in vivo double labeling, they are able to separate neural crest cells before and after their migration and differentiation at different stages to compare them with non-neural crest cells. This comparison is based on studying those microRNAs that are dysregulated in neural crest cells compared to other cells. Although the article is well-structured and based on strong evidence supported by clear and detailed experimental work, some considerations should be taken into account, before it can be further considered for publication

Major revisions:

1. The authors should analyze whether the microRNAs that are upregulated at 16 hpf-miR-133a, miR199-2, and miR-338-remain upregulated also at 28 hpf stage.

2. In section 2.2 of Results, the authors refer to various mRNAs, such as Sox9b, indicating that in previous studies conducted by other laboratories, these genes display opposite expression patterns as compared to the microRNAs analyzed on this study. However, they have not carried out this verification themselves. They should analyze the expression of these mRNAs by qPCR to support that similar opposite expression between the miRNAs and mRNAs is indeed observed in this biological context.

Furthermore, they do not mention whether the expression of these microRNAs occurs in the entire embryo or in the cranial part. Given that the expression of microRNA 133a, for example, is not exclusively cranial, they should indicate whether such increase during development may indicate, or not, a relevant role in the development and differentiation of neural crest-derived cells.

3. The authors do not explain the possible role of miR-338 in melanophore differentiation. In that same section, they mention key genes in this process, such as *Kita*, *Sox10*, and *tyrp1a*. Why did not the authors study the expression of these genes by qPCR in miR338 overexpression? To my point of view, this is compulsory.

Minor revisions:

1. The results section should be reworked as it contains a lot of information that should be included in the Discussion section. Many sentences in this section explain discrepancies in the results compared to other studies. This type of explanation and discussion of the results with previous literature should be addressed in the Discussion section.

2. The miR133a section does not mention its role as a regulator in developmental processes and its role in cell progenitor proliferation. Therefore, it is suggested to the authors to include the following references:

Sharma G, Mo JS, Lamichhane S, Chae SC. MicroRNA 133A Regulates Cell Proliferation, Cell Migration, and Apoptosis in Colorectal Cancer by Suppressing CDH3 Expression. *J Cancer*. 2023 Apr 9;14(6):881-894. doi: 10.7150/jca.82916. PMID: 37151391; PMCID: PMC10158507.

Ivey KN, Srivastava D. MicroRNAs as regulators of differentiation and cell fate decisions. *Cell Stem Cell*. 2010 Jul 2;7(1):36-41. doi: 10.1016/j.stem.2010.06.012. PMID: 20621048.

Garcia-Padilla C, Garcia-Lopez V, Aranega A, Franco D, Garcia-Martinez V, Lopez-Sanchez C. Inhibition of RhoA and Cdc42 by miR-133a Modulates Retinoic Acid Signaling During Early Development of Posterior Cardiac Tube Segment. *Int J Mol Sci*. 2022 Apr 10;23(8):4179. doi:10.3390/ijms23084179. PMID: 35456995; PMCID: PMC9025022.

Manuscript ID: LSA-2025-03431R

Title: miR-133a-3p and miR-338-3p Shape Neural Crest Derivatives in Zebrafish

Dear Editor and Reviewers,

We would like to thank the Reviewers for their careful evaluation of our manuscript and for their highly constructive comments. We agree that the requested controls and validations were critical, and we greatly appreciate the opportunity to strengthen our study.

In accordance with the Reviewers' suggestions, we have made substantial improvements to the manuscript. Most notably, we have:

- Provided comprehensive validation of our FACS NCC-isolation strategy, including representative gating plots and RT-qPCR confirmation of sorted population purity.
- Performed new *in vivo* RT-qPCR experiments to validate the expression of key melanophore differentiation genes (*kita* and *tyrp1a*) upon miR-338-3p overexpression.
- Extensively revised the text to properly contextualize our gain-of-function approach, carefully framing our findings in terms of sufficiency and highlighting our internal and methodological controls.

We believe that these new experiments and textual revisions have significantly elevated the rigor and clarity of our work. Additionally, for clarity and consistency with miRNA naming conventions, the nomenclature for the studied miRNAs has been updated throughout the text to specify the mature strand (the 3p strand for miR-133a, miR-199, and miR-338; and the 5p strand for miR-145). The manuscript title has also been updated to reflect this change.

Below, please find our detailed, point-by-point responses to each comment. For clarity, the Reviewers' original comments are shown in bold, and our responses are provided in standard text. For reference, you will also find the original text with the changes highlighted at the end of this document. The new, clean full manuscript is submitted as a different document.

Reviewer #1 (Comments to the Authors (Required)):

In this manuscript, the authors investigate the role of microRNAs (miRNAs) in neural crest cells (NCCs) using zebrafish as a model organism. To this end, the authors isolated NCCs and non-NCCs by fluorescence-activated cell sorting (FACS), followed by RNA-Seq analysis at two developmental stages (16 hpf and 18 hpf). Based on miRNA enrichment in the NCC population and subsequent gene network analyses of putative target genes, the authors focused on miR-133a, miR-199-2, and miR-388 for functional studies. Overexpression of each miRNA, alone or in combination, resulted in phenotypes affecting pigment cells and craniofacial cartilage development. The authors also identified several mRNA targets of these miRNAs through computational prediction and reporter assays.

While the identification of these three miRNAs as potential regulators of NCC development is of interest, the manuscript is not well organized and is often difficult to follow. The data presented are generally of good quality, but several experiments lack appropriate controls and validation steps.

In this reviewer's opinion, the manuscript is currently premature and requires significant reorganization and clarification before it can be considered a solid research article.

Major points

1. The authors isolated NCCs and non-NCCs by FACS from Tg(*sox10:eGFP,sox10:mRFP*) embryos. However, validation of this approach, which is critical to the subsequent analysis, was not provided. First, a representative FACS plot should be shown. Second, the purity of the sorted NCC and non-NCC populations should be validated, for example, by RT-PCR of established marker genes for neural crest and other cell types.

Expected timeframe: one-two months.

We agree that validating the FACS strategy and the purity of the sorted populations is essential. We have now expanded the manuscript to explicitly document (i) the gating strategy with representative plots and (ii) enrichment/depletion of established marker genes in the sorted fractions.

Representative FACS plots and gating strategy:

We now include representative flow cytometry/ plots showing the sequential gating used to isolate NCCs from Tg(*sox10:eGFP, sox10:mRFP*) embryos (debris exclusion by FSC/SSC, singlet gating, and fluorescence gates). These plots have been added as Fig. S1A. We also clarified in the Methods (Section 4.2) that wild-type embryos were used to set the negative fluorescence gate and Tg(*actb1:eGFP*) embryos were used as a positive control for instrument settings and gating, as described.

Validation of sorted population purity by marker gene expression:

To verify that the GFP+/RFP+ fraction corresponds to NCCs and that the GFP- fraction is depleted of NCCs, we performed RT-qPCR on RNA extracted after sorting. As NCC markers we used *foxd3*, while as a non-NCC control we used *tbx5a* (lateral plate mesoderm) as indicated in the revised manuscript. The NCC fraction showed strong enrichment of NCC markers and depletion of the non-NCC marker, whereas the non-NCC fraction showed the reciprocal pattern. These validation data are now presented in Fig. S1D and are referenced in the Methods sections (4.2).

In addition, we now explicitly state in the Methods that within the eGFP+ gate, >99% of cells were mRFP+, supporting the specificity of the sorted NCC population (revised Section 4.2).

2. The rationale for selecting the three miRNAs for functional analysis is not clearly or systematically presented. In the Results section, each miRNA is preceded by a lengthy introductory paragraph summarizing previous findings; however, much of this content is descriptive and not essential to justifying their selection. A more concise and focused explanation of why these particular miRNAs were chosen would improve clarity.

We agree and thank the reviewer for this suggestion. In the revised manuscript we have streamlined the presentation of candidate selection and made the prioritization criteria explicit.

Specifically, we now provide a concise, systematic rationale in Results Section 2.1/2.2 summarizing the key criteria used to select miR-133a-3p, miR-199-3p, and miR-338-3p for functional analysis:

- NCC enrichment at the premigratory stage: all three miRNAs were significantly overrepresented in the 16 hpf FACS-isolated *sox10+* population (RNA-seq), and this enrichment was independently validated by stem-loop RT-qPCR on sorted cells.
- Developmental timing: enrichment was observed at 16 hpf, prior to migration and differentiation, consistent with potential roles in early NCC programs.
- Predicted functional relevance to NCC biology: *in silico* target prediction and network/enrichment analyses identified predicted targets and interaction nodes linked to processes central to NCC development (e.g., EMT/migration, proliferation, and differentiation; including NCC-relevant regulators such as *sox9b*, *foxd3*, *snai2* in the predicted target networks).
- Prior association with analogous cellular programs in cancer: these miRNAs have been reported as tumour-suppressive regulators of EMT, migration, and proliferation in multiple cancer contexts, processes that are mechanistically related to NCC behaviour, which motivated testing their developmental roles *in vivo*.

To improve readability, we relocated much of the miRNA-by-miRNA descriptive background from the beginning of each Results subsection, as a summary table (Supplementary Table ST3) and in the Discussion Section.

3. The functional analysis of the three miRNAs relies primarily on overexpression experiments, which are known to potentially induce artificial interactions between miRNAs and mRNAs containing seed-matched sites. To strengthen the conclusions drawn from these experiments, complementary loss-of-function approaches-such as genetic mutants or antisense oligonucleotides-are necessary to validate the physiological relevance of the observed effects.

Expected timeframe: three-six months.

We agree with the reviewer that loss-of-function experiments would provide important complementary evidence and would further strengthen the physiological interpretation of our gain-of-function data. In the current study, our primary aim was to test whether the overexpression of miRNAs enriched in the NCC are sufficient to perturb NCC-derived developmental programs and to identify direct regulatory links to candidate targets.

We attempted to implement miRNA inhibition using sponge constructs; however, due to technical difficulties in cloning repetitive tandem binding-site arrays, we were not able to generate validated sponge mRNAs within the current revision timeframe due to the scale of reagent development and validation required. Establishing genetic mutants (e.g., CRISPR/Cas9 deletion of miRNA loci) or optimizing antisense/LNA inhibitors also requires substantial additional time for design, validation of knockdown efficiency, and phenotypic analysis, and is therefore beyond the scope of the present revision.

To mitigate concerns inherent to overexpression-based approaches, we have revised the manuscript to adapt wording where appropriate (i.e., framing phenotypes as demonstrating sufficiency rather than necessity), and emphasize multiple lines of evidence supporting specificity and developmental relevance:

- Experimental controls: dsRED-only injections control for injection/transcript effects, and miR-199-3p (also NCC-enriched) does not produce the same phenotypes under identical injection conditions, arguing against nonspecific toxicity or global disruption of the miRNA machinery. This has been added to the Discussion section.
- Mechanistic specificity: reporter assays demonstrate direct repression via the sox9b 3'UTR, supporting a specific miRNA-target interaction rather than solely artificial seed-mediated effects.

4. Related to the above point, the choice of control in overexpression experiments is also critical. Using a control mRNA lacking a miRNA sequence is insufficient. A more appropriate control would be the overexpression of an unrelated miRNA or miRNA harboring mutations in the seed region, to account for potential non-specific effects. Overexpression can lead to saturation of Argonaute (Ago) proteins and induce off-target phenotypes, as previously reported (PMID: 21576259). Careful experimental design is required to distinguish specific miRNA function from such artifacts.

Expected timeframe: two-three months.

We agree with the reviewer that appropriate controls are essential for interpreting miRNA gain-of-function experiments and for addressing nonspecific effects such as Ago saturation and off-target repression (PMID: 21576259). We have revised the manuscript to clarify our control strategy and to discuss this limitation explicitly.

All pri-miRNA overexpression constructs (miR-133a-3p, miR-199-3p, and miR-338-3p) were cloned into the same dsRED expression vector backbone and were microinjected at the same total transcript dose, with a dsRED-only transcript used to control for injection and transcript load. Although miR-199-3p was included as a candidate miRNA rather than selected *a priori* as an “unrelated” control, it provides an important internal comparator: under identical vector and dosing conditions, miR-199-3p overexpression did not produce detectable craniofacial cartilage, in contrast to miR-133a-3p and miR-338-3p. This argues against the miR-133a-3p/miR-338-3p phenotypes being solely due to generalized overexpression artifacts (including uniform saturation of the miRNA machinery).

We agree that a seed-mutant miRNA would be an additional stringent specificity control. However, generating and validating seed-mutant pri-miRNA constructs (including confirming correct processing into mature miRNA and comparable expression levels) would require additional time and is beyond the scope of the current revision; we now note this as a future direction.

5. Regarding the phenotypic analysis, bright-field images of whole embryos at lower magnification are warranted to assess overall body morphology. Such views would help determine whether the observed phenotypes are specific or part of broader developmental defects, and would strengthen the interpretation of the results.

Expected timeframe: one month.

We thank the reviewer for this suggestion and agree that assessing general morphology is crucial to distinguish specific phenotypic effects from broader developmental delays or toxicity.

As requested, we acquired low-magnification bright-field images of whole embryos at ~16 hpf and larvae at 48 hpf across all experimental conditions (Control, miR-133a, miR-199, miR-338, and co-injection). To provide a quantitative assessment of developmental progression, we measured somite number at 16 hpf and eye surface area at both 16 and 48 hpf.

These analyses reveal that there were no significant differences in somite number or overall body size compared to controls, indicating that the observed NC defects are specific and not secondary to general developmental retardation. These images and quantifications are now included in Figure S5E-Q.

6. In Figure 7G, the reporter analysis lacks an internal control for quantification. Injection-based assays are inherently less precise than techniques such as micropipetting, making normalization essential. To ensure reliable quantification-especially for the subtle effects typically observed

with miRNA-mediated regulation-it is necessary to include an internal control, such as a co-injected mRNA encoding an independent reporter protein.

Expected timeframe: one-two months.

We agree that a dual-reporter, internally normalized design is ideal for injection-based miRNA sensor assays. In our experiments, the readout was based on a single fluorescent reporter (*d4GFPn*) fused to different 3'UTRs, and thus we cannot add a second, independently quantifiable reporter retrospectively for ratiometric normalization in the current revision.

We now explicitly state that the “control-3'UTR” reporter uses the *Xenopus* β -*globin* 3'UTR (a heterologous 3'UTR commonly used to provide stable expression and not to containing functional binding sites for the tested zebrafish miRNAs). This construct should therefore serve as an appropriate negative control for assessing miRNA-specific repression mediated by the *sox9b* 3'UTR, rather than effects on the reporter coding sequence or general translation. The two *sox9b* 3'UTR sensors display the expected differential responsiveness based on their binding-site composition (miR-133a-3p represses only UTR-1, whereas miR-338-3p represses both UTR-1 and UTR-2), supporting sequence-dependent regulation rather than nonspecific injection effects.

Although our pri-miRNA overexpression constructs include dsRED, dsRED fluorescence is not an unbiased injection/loading control in this setting because the same transcript contains the pri-miRNA hairpin that is processed by the Microprocessor pathway, potentially affecting transcript stability and reporter output in a miRNA-dependent manner.

In addition, we now explicitly state that the reporter assay was performed in two independent experiments, which yielded consistent results; for clarity, one representative experiment is shown in Fig. 7, and the replicate is provided as supplementary figure (Fig S6I-K).

7. To further support the direct interaction between miRNAs and their target mRNAs, reporter assays using constructs with mutated target sites would be desirable. Including such experiments would help demonstrate target site specificity and strengthen the overall interpretation.

Expected timeframe: one-two months.

We agree that mutation of the predicted binding sites in the *sox9b* 3'UTR sensors would be a stringent additional test of site-level specificity. However, generating and validating mutated reporter constructs is beyond the scope of the current revision. As discussed in our previous response, we instead emphasize the built-in specificity controls already present in our sensor design (control 3'UTR and differential responsiveness of *sox9b*-UTR-1 versus *sox9b*-UTR-2 consistent with predicted site composition) and we have tempered the corresponding conclusions in the text.

8. The figures lack figure number labeling, which significantly disrupts the flow of reading and makes the manuscript difficult to follow.

We apologize for this oversight. We have revised all main and supplementary figures to include explicit figure numbers on the figure files.

Minor points

1. On page 2, line 72, "The miRISC complex targets mRNA by binding to a specific "seed region" of the miRNAs" This is incorrect. The miRISC complex targets mRNAs through the binding of a specific "seed region" of the miRNA to complementary sites within the mRNA.

We thank the reviewer for noting this. We have revised the text to clarify this point.

2. On Page 5, line 200, the statement "Since the expression profiles of miR-133a, miR-199-2, and miR-338-3p in zebrafish remain incompletely characterized" is inaccurate. Even early studies on miRNA profiling in zebrafish, such as the one by Wienholds et al. (PMID: 15937218), reported temporal expression data for miR-133 and miR-199. Furthermore, numerous follow-up studies have since provided detailed expression profiles of these miRNAs. The authors should revise this statement to accurately reflect the existing literature.

We thank the reviewer for this comment. We have revised the sentence to acknowledge this literature and to more precisely state our motivation: to provide quantitative RT-qPCR measurements for the specific mature miRNAs examined here (miR-133a-3p, miR-199-3p, and miR-338-3p) across the stages analysed in this study.

3. In several instances, the authors appear to conflate miRNA targets with target candidates or putative targets based solely on prediction. For example, on page 6, line 216, the phrase "Several of the identified target genes have been..." is misleading, as the discussion refers to predicted candidates rather than experimentally validated targets. It is important to clearly distinguish between computational predictions and experimentally confirmed miRNA-mRNA interactions, as the latter requires direct validation.

In addition, the manuscript inconsistently refers to translational repression in contexts where the observed effects may reflect a combination of translational repression and mRNA decay. While the authors briefly acknowledge this distinction, their terminology is not used consistently throughout the manuscript. To avoid overinterpretation, the authors should revise these statements for clarity and accuracy.

We thank the reviewer for pointing this out. We agree that computationally predicted miRNA-mRNA interactions should be clearly distinguished from experimentally validated targets. We have revised the manuscript to consistently refer to these genes as "predicted/putative targets" or "target candidates" unless direct validation is provided (e.g., the *sox9b* 3'UTR reporter assay). In addition, we have standardized our terminology regarding miRNA mechanism by using "post-transcriptional repression" as the general term and, where appropriate, specifying that our observations are consistent with mRNA destabilization and/or translational repression rather than implying one mechanism exclusively.

4. On page 6, line 245" This aligns with the role of miRNAs as fine tuners of gene expression, where their regulatory impact does not necessarily scale linearly with increased expression." Citation is required for this specific statement.

We thank the reviewer for this suggestion. We have added appropriate citations to support this statement (e.g., Bartel, 2009; Kloosterman & Plasterk, 2006) and slightly revised the wording to avoid over-specificity.

5. On page 8, line 307, the term "d4GFPn" is introduced without explanation. All specific terms, constructs, or materials should be clearly defined at their first appearance to ensure clarity for readers who may not be familiar with them.

We thank the reviewer for noting this. We have revised the text to define d4GFPn at its first appearance (nuclear-localized destabilized eGFP with an ~4 h half-life) and ensured consistent nomenclature throughout the manuscript.

Reviewer #2 (Comments to the Authors (Required)):

The work by Steeman et al. investigates the role of microRNAs as modulators of neural crest cell lineages. Using in vivo double labeling, they are able to separate neural crest cells before and after their migration and differentiation at different stages to compare them with non-neural crest cells. This comparison is based on studying those microRNAs that are dysregulated in neural crest cells compared to other cells. Although the article is well-structured and based on strong evidence supported by clear and detailed experimental work, some considerations should be taken into account, before it can be further considered for publication

Major revisions:

1. The authors should analyze whether the microRNAs that are upregulated at 16 hpf-miR-133a, miR199-2, and miR-338-remain upregulated also at 28 hpf stage.

We thank the reviewer for raising this point. We have already profiled miRNA expression in FACS-isolated NCC and non-NCC populations at both 16 hpf and 28 hpf by small RNA-seq (Fig. 1A; Table ST2). We have revised the Results text to more explicitly report the 28 hpf NCC/non-NCC enrichment values for miR-133a-3p, miR-199-3p, and miR-338-3p and to clarify the conclusion: these miRNAs are enriched in NCC at 16 hpf, but are no longer enriched (and/or are reduced/absent in NCC) at 28 hpf compared to 16 hpf, consistent with a predominant role during early/premigratory stages.

2. In section 2.2 of Results, the authors refer to various mRNAs, such as Sox9b, indicating that in previous studies conducted by other laboratories, these genes display opposite expression patterns as compared to the microRNAs analyzed on this study. However, they have not carried out this verification themselves. They should analyze the expression of these mRNAs by qPCR to support that similar opposite expression between the miRNAs and mRNAs is indeed observed in this biological context. Furthermore, they do not mention whether the expression of these

microRNAs occurs in the entire embryo or in the cranial part. Given that the expression of microRNA 133a, for example, is not exclusively cranial, they should indicate whether such increase during development may indicate, or not, a relevant role in the development and differentiation of neural crest-derived cells.

We thank the reviewer for this suggestion. We agree that the manuscript should clearly distinguish between our experimental data and expression patterns reported in published datasets. In the revised manuscript, we have removed/relocated statements implying that we directly observed inverse expression between these miRNAs and candidate mRNAs, and we now present the mRNA expression patterns strictly as contextual information from published zebrafish expression resources (with appropriate citations), discussed in the Discussion rather than the Results.

We now explicitly state in Section 2.3 that the developmental time-course RT-qPCR (Fig. 3) was performed on whole-embryo extracts and therefore reflects global expression rather than cranial- or NCC-restricted expression. We also expanded the discussion to note that miR-133a is known to become strongly expressed in non-NC tissues (e.g., muscle) later in development, which likely contributes to the whole-embryo increase at later stages.

3. The authors do not explain the possible role of miR-338 in melanophore differentiation. In that same section, they mention key genes in this process, such as *Kita*, *Sox10*, and *tyrp1a*. Why did not the authors study the expression of these genes by qPCR in miR338 overexpression? To my point of view, this is compulsory.

We thank the reviewer for this suggestion. These analyses were in fact performed and are included in the new manuscript version. In miR-338 overexpression embryos, we measured melanophore regulatory/differentiation genes by RT-qPCR at 24 hpf. Specifically, *kita* and *tyrp1a* mRNA levels are significantly reduced upon miR-338 overexpression (Fig. 4G-H), consistent with impaired melanophore development. Regarding *sox10*, it was shown in the craniofacial development experiments (Fig. 6D), and we observed that *sox10* mRNA is reduced primarily upon miR-133a overexpression, not miR-338 overexpression. We have now cleared this up and expanded the Discussion of how miR-338 may influence melanophore differentiation via effects on *kita/tyrp1a*, rather than via *sox10* downregulation.

Minor revisions:

1. The results section should be reworked as it contains a lot of information that should be included in the Discussion section. Many sentences in this section explain discrepancies in the results compared to other studies. This type of explanation and discussion of the results with previous literature should be addressed in the Discussion section.

We thank the reviewer and agree. We have streamlined the Results section to focus on our experimental findings and moved literature comparisons and interpretation of discrepancies primarily to the Discussion. This improves clarity and keeps the Results more descriptive.

2. The miR133a section does not mention its role as a regulator in developmental processes and its role in cell progenitor proliferation. Therefore, it is suggested to the authors to include the following references:

Sharma G, Mo JS, Lamichhane S, Chae SC. MicroRNA 133A Regulates Cell Proliferation, Cell Migration, and Apoptosis in Colorectal Cancer by Suppressing CDH3 Expression. J Cancer. 2023 Apr 9;14(6):881-894. doi: 10.7150/jca.82916. PMID: 37151391; PMCID: PMC10158507.

Ivey KN, Srivastava D. MicroRNAs as regulators of differentiation and cell fate decisions. Cell Stem Cell. 2010 Jul 2;7(1):36-41. doi: 10.1016/j.stem.2010.06.012. PMID: 20621048.

Garcia-Padilla C, Garcia-Lopez V, Aranega A, Franco D, Garcia-Martinez V, Lopez-Sanchez C. Inhibition of RhoA and Cdc42 by miR-133a Modulates Retinoic Acid Signaling During Early Development of Posterior Cardiac Tube Segment. Int J Mol Sci. 2022 Apr 10;23(8):4179. doi:10.3390/ijms23084179. PMID: 35456995; PMCID: PMC9025022.

We thank the reviewer for these helpful references. We have incorporated them into the Introduction/Discussion when describing known roles of miR-133a in regulating differentiation and cell fate decisions and its broader links to proliferation/migration/apoptosis programs, and we cite the suggested papers accordingly.

miR-133a-3p and miR-338-3p Shape Neural Crest Derivatives in Zebrafish

Abstract

Neural crest cells are a transient, multipotent population that gives rise to diverse structures during vertebrate embryonic development, including craniofacial cartilage and pigment cells. While the transcriptional regulation of neural crest development is well characterized, the role of microRNAs remains less understood. Using a double-transgenic zebrafish model expressing fluorescent reporters under the *sox10* promoter, combined with fluorescence-activated cell sorting and RNA sequencing, we identified microRNAs enriched in neural crest cells. We focused on miR-133a-3p and miR-338-3p, previously linked to tumor suppression, to explore their developmental roles. Overexpression of either microRNA led to craniofacial cartilage malformations and reduced melanophore number, accompanied by decreased expression of key regulators including *sox9b*, *sox10*, and *runx3*. Reporter assays confirmed direct targeting of the *sox9b* 3' untranslated region. Additionally, miR-338-3p overexpression increased neural crest cell numbers, suggesting a role in proliferation. These findings uncover novel functions for miR-133a-3p and miR-338-3p in vertebrate craniofacial and pigment cell development, highlighting shared regulatory features between embryogenesis and tumorigenesis.

Introduction

During vertebrate embryogenesis, the development of organs depends on the coordinated formation, remodeling, and, in some cases, the regression of various tissues and structures. Among these, the neural crest (NC) is a transient, multipotent cell population that emerges early in development and plays a pivotal role in generating diverse cell lineages. Initially located dorsally to the neural tube, NC cells (NCC) undergo extensive proliferation followed by epithelial-to-mesenchymal transition (EMT), which facilitates their delamination and migration (Thiery et al, 2009; Theveneau & Mayor, 2012). Subsequently, these cells differentiate into a variety of derivatives, including neurons, glial and Schwann cells, craniofacial chondrocytes, pigment cells, and endocrine cells (Dupin & Le Douarin, 2014). Defects in NC development are linked to neurocristopathies and certain cancers. Thus, elucidating the molecular mechanisms governing NC formation, migration, and differentiation is critical for understanding the etiology of these disorders and for advancing therapeutic approaches (Vega-Lopez et al, 2018; Weiner et al, 2020). NC development is regulated by a complex network of transcription factors (TFs) and signaling molecules, organized in a gene regulatory network (GRN) (Martik & Bronner, 2017) that operates in a spatiotemporal manner. In zebrafish, NC formation is initially guided by extrinsic factors, such as the BMP and WNT pathways (Hammerschmidt et al, 1996; Schumacher et al, 2011), which activate NC-specific genes such as *foxd3*, *snai1/2*, and *sox10* (Lewis et al, 2004). During EMT, the expression of new TFs such as *twist1a/b* facilitates the switch from epithelial to mesenchymal cadherins, among other changes, enabling migration (Cano et al, 2000; Ferronha et al, 2013; Martik & Bronner, 2017). Once NCC reach their destination, additional external signals induce further regulatory changes specific to the GRN of each derivative.

NCC give rise to pigment cells, such as melanophores, iridophores and xanthophores in zebrafish. Melanophores are responsible for black pigmentation and emerge at 24 hours post-fertilization (hpf); their appearance depends on the expression of *sox10* and *pax3/7*, which activate the master gene *mitfa* (Kelsh et al, 1996). *Mitfa*, in turn, induces the expression of melanogenic genes such as *dct*, *tyr*, and *pmel* (Bharti et al, 2006; Greenhill et al, 2011).

Epigenetic regulators like HDAC1 appear to repress alternative NC fates (Greenhill et al, 2011), favoring melanophores differentiation. Iridophores, responsible for iridescence through guanine crystals, appear at around 72 hpf (Kelsh et al, 1996; Gur et al, 2020) and follow a similar regulatory pathway but diverges through Tfec-mediated activation of TFs *ltk* and *sox10* (Lopes et al, 2008; Kaller & Hermeking, 2016; Petratou et al, 2018), among other factors. NCC also contribute to craniofacial chondrocytes development (Kimmel et al, 2001; Glasauer & Neuhaus, 2014) and skull formation in zebrafish. In this process, *sox9* paralogs (*sox9a* and *sox9b*) (Chiang et al, 2001; Yan et al, 2005), modulated by *runx3* and *egr3*, regulate chondrocyte stacking and proliferation (Dalcq et al, 2012). Sox9a/b, in conjunction with Sox5/6, activate the expression of *col2a1a*, a key collagen-coding gene in cartilage, and *agc1*, a marker of cartilage differentiation. Given the complexity of craniofacial development, disruptions in these regulatory pathways can contribute to congenital disorders.

Recent studies have highlighted the critical role of microRNAs (miRNAs) in NC development (Wienholds et al, 2003; Weiner, 2018). These 22nt endogenous, non-coding RNAs regulate mRNA stability and transcription (Kloosterman et al, 2006) through the miRISC complex, which includes the RNAse Argonaute. The miRISC complex targets mRNA through the binding of by binding to a specific “seed region” of the miRNAs to complementary sites within the mRNA (Lewis et al, 2005; Bartel, 2009), typically within the mRNA’s 3’UTR, and each miRNA family has an average of 300 binding sites under selective pressure (Stark et al, 2005; Friedman et al, 2009). As a result, a single mRNA can be regulated by multiple miRNAs, while each miRNA can regulate numerous genes. miRNAs-MicroRNAs are indispensable in early embryonic development, influencing critical events such as brain morphogenesis and maternal mRNA clearance (Giraldez et al, 2005, 2006). In the context of NC development, miRNAs dysregulation has been implicated in neurocristopathies, including DiGeorge syndrome and congenital central hypoventilation syndrome (Wienholds et al, 2003; Antonaci & Wheeler, 2022; Bamforth & Burn, 2023). Furthermore, miRNAs are often dysregulated in tumors, acting as either oncogenic (*oncomiRs*) or tumor-suppressive (*anti-oncomiRs*) regulators (Svoronos et al, 2016; Syeda et al, 2020). NC-derived cancers, such as neuroblastomas and melanomas, share molecular features with developmental processes, highlighting the need for further exploration into the roles of miRNAs in both NC development and cancer progression (Antonaci & Wheeler, 2022).

In this study, we aim to identify and characterize miRNAs implicated in NC development through gain-of-function experiments in zebrafish embryos. By leveraging a double-transgenic zebrafish model (Tg(*sox10:eGFP*, *sox10:mRFP*)), we isolated NC and non-NC populations via fluorescence-activated cell sorting (FACS) and conducted transcriptomic analyses. Our findings reveal that some specific miRNAs regulate key genes involved in NC-derived structures, particularly craniofacial cartilage and melanophores, through interactions with transcription factors such as Sox9b and Runx3. Notably, these miRNAs share regulatory mechanisms with tumor suppressor pathways, suggesting a broader role in cellular plasticity and differentiation. This research advances our understanding of vertebrate embryogenesis and provides valuable insights into the molecular mechanisms underlying neurocristopathies, NC-derived cancers, and tumorigenesis, underscoring the need for further investigation into the regulatory functions of miRNAs in both developmental and pathological contexts.

Results

miR-133a, miR-199-2 and miR-338-3p are overrepresented in NCC.

Using the double transgenic zebrafish line Tg(*sox10*:eGFP,*sox10*:mRFP) (Kwak et al, 2013), we separated the NCC and non-NCC population with FACS from 16 hpf (before the NCC migration and differentiation of NC derivatives) and 28 hpf embryos (after NCC migration and during differentiation of derivatives) and performed RNA-seq on each subpopulation.

At 16 hpf, we identified a total of 225 miRNAs, with 60 and 57 being miRNAs specific to NCC and non-NCC populations, respectively (Fig. 1A x-axis; Fold Enrichment > 2, Table ST2, Fig. S1B). In the NCC subpopulation, we found several previously reported NCC-associated miRNAs, such as miR-96 and miR-204 (Gessert et al, 2010; Avellino et al, 2013). At 28 hpf, we identified 50 miRNAs in NCC, of which 12 miRNAs were overrepresented compared to non-NCC and one uniquely present in this subpopulation (miR-24, Fig. 1A y-axis, Fig. S1C). Conversely, the non-NCC population at 28 hpf contained a total of 187 miRNAs, with 138 unique to this subpopulation and 17 overrepresented compared to NCC. Only three miRNAs were overrepresented in both timepoints of the NCC (miR-125-5p, miR-16b and miR-462), previously reported in neural or neural crest development, immune response and hematopoiesis (Boissart et al, 2012; Singh et al, 2015; Tian et al, 2016; Huang et al, 2019; Antonaci & Wheeler, 2022) (circled in Fig. 1A, Table ST2).

Further analysis was performed on miRNAs over-represented in NCC at the 16 hpf stage in zebrafish, prior to NCC differentiation and migration. ~~This early developmental time point was selected because it allows for the identification of~~ We focused on this early time point to identify miRNAs potentially involved in the initial specification and commitment of NCC, ~~minimizing confounding effects from downstream lineage-specific expression.~~ Based on bioinformatics analyses including STRING protein-protein interaction networks, Gene Ontology (GO) classifications and KEGG pathway enrichments (Fig. 2, Fig. S1, S2, S3), we identified miR-133a-3p, miR-199-3p2, and miR-338-3p as key candidates for further investigation. These candidates were prioritized because they had been reported as tumor suppressors in cancer models, where they regulate EMT, cell migration, and proliferation, processes analogous to those in NCC development (summarized in Table ST3). Among these candidates, miR-199-3p and miR-338-3p were absent, while miR-133a-3p was underrepresented in the 28 hpf NCC population (Fig. 1A, Fig. S1B, C). ~~Besides, a literature review (see below) supported their potential roles in NCC specification and cancer biology.~~

Stem-loop RT-qPCR was performed on total RNA extracted from freshly dissociated, fluorescently sorted cells of Tg(*sox10*:eGFP, *sox10*:mRFP) (Kwak et al, 2013) embryos at 16 hpf. miRNAs miR-133a, miR-199-2, and miR-338-3p were found to be overexpressed in NCC (Fig. 1B), consistent with RNA-seq data. As a non-NCC control, we analyzed miR-145 expression (Fig. 1B), showing similar expression levels in NCC and non-NCC populations, as observed in the RNA-seq results (Fig. 1A, C, black dot).

In silico target prediction and functional enrichment analysis

To identify potential targets of the candidate miRNAs in NCC development, we performed computational predictions followed by functional enrichment analysis. Results are briefly described below.

miR-133a

~~miR-133a was initially identified in muscle tissue of zebrafish embryos at 48 hpf (Chen et al, 2005; Wienholds et al, 2005), and observed in the mouse heart and cerebral cortex (Lagos-Quintana et al, 2002). It has been proposed as a biomarker for myocardial infarction due to decreased expression under this condition (Boštjančič et al, 2010). miR-133a suppresses the~~

gene *Coll1a1* in myocardial fibrosis (Castoldi et al, 2012) and aids in cardiac tissue regeneration (Kimmel et al, 2001; Yin et al, 2012b). In mice, miR-133a is co-transcribed with miR-1, both regulating processes in skeletal muscle development (Koutsoulidou et al, 2011). In cancer, miR-133a functions as a tumor suppressor by modulating FSCN1 expression in breast cancer (Wu et al, 2012) and, along with miR-145, in bladder and esophageal cancers. Together with miR-1, it regulates proteins like TAGLN2 and PNP, inhibiting tumor proliferation in renal, bladder, prostate, and rhabdomyosarcoma cancers, and helping to prevent myocardial ischemia (Rao et al, 2010; He et al, 2011; Yoshino et al, 2011; Kawakami et al, 2012; Kojima et al, 2012). Circular RNAs circFUT10 and circ_nxcl inhibit miR-133a, promoting mioblastomas and myocardial ischemia, respectively (Li et al, 2018a, 2018b). Although these studies were conducted in various model systems and pathological contexts, they collectively support a broader role for miR-133a in regulating cell cycle progression and cytoskeletal organization.

We retrieved a total of 2966 genes containing miR-133a-3p putative binding sites from the TargetScan database (context+ score+ between -0.01 and -1.05, Supplementary Table S3S4). According to g:Profiler ($P_{adj} < 0.05$, g:SCS Threshold), multiple terms associated with neuronal development were overrepresented (Fig. 2A, Table S4S5). Additional terms, such as "apoptosis apoptotic signaling pathways," "anatomical structure development," and "developmental processes" were also identified, though neuronal development terms were the most predominant. Similar results were obtained using Panther (p -value < 0.05 , FDR test), which also highlighted terms like "angiogenesis" and "axon development guidance." DAVID (p -value < 0.05 , FDR test) revealed NC-related terms, such as "neural crest cell migration" and "cartilage development." No NC development process pathway was overrepresented in this gene list (KEGG). Using the bioinformatic tool STRING, which generates interaction networks from gene lists, we identified central node genes that participate in multiple processes or interact with several genes within the list. This approach allowed for a focused analysis of miR-133a-3p's impact on genes that may have a significant influence on zebrafish embryonic development. By analyzing a gene list with at least one putative miR-133a binding site and expression data from ZFIN for NCC or their derivatives, other central node genes were elucidated, including *ddx21*, *pax9*, *sox10*, *erbb2*, and *gf20b* (Fig. 2B, Fig. S2).

miR-199-2

miR-199-2 has been identified in zebrafish development (starting from 20 hpf, primarily in the fins) as well as in various other vertebrates (Lim et al, 2003; Wienholds et al, 2005). It has been shown to regulate ITGA3 expression (involved in cell adhesion), inhibiting cell migration and tumour invasion in squamous cell carcinoma (Koshizuka et al, 2017). In liver cancer cell cultures, miR-199-2 inhibits EMT (Zhang et al, 2019) and regulates apoptosis in intervertebral disc cells by inhibiting MAP3K5 (Wang et al, 2018). These findings position miR-199-2 as a tumour suppressor miRNA, like miR-133a. Its inhibitory effect on EMT in cancer may also be relevant to NC development and cell migration.

For GO and KEGG pathway analysis, we extracted a list of 3,000 genes with predicted binding sites for miR-199-2-3p from TargetScan, with context+ scores between -0.04 and -9.1, although a few overrepresented terms were found (Fig. S3A, Tables S5 S6 and S6S7). According to g:Profiler, 7 of 19 terms were related to nervous system development, while others were more general terms such as "multicellular organism development" and "developmental processes". Comparable results were obtained with Panther and DAVID, with terms like "axonogenesis" and "tissue development." A single KEGG pathway, "ErbB signaling pathway," was overrepresented. The absence of terms related to the NC, or its derivatives does not necessarily indicate that miR-199-2-3p is uninvolved in NC development,

as it may regulate only a few key genes essential for this process. STRING analysis, using genes with putative binding sites for miR-199-2 and expression reported in ZFIN for NCC or their derivatives, identified several central node genes, including *foxd3*, *wnt5b*, *erbb2*, and various genes in the FGF signaling pathway (Fig S2S3B).

miR-338-3p

~~Bibliographic analysis revealed that miR-338-3p is linked to the nervous system, having been detected in neuron polyribosomes (Kim et al, 2004) and promoting nervous system development (Chen et al, 2013; Howe et al, 2017). miR-338-3p inhibits EMT in gastric cancer by regulating ZEB2 (Huang et al, 2015), suppresses OSCC (oral cancer) cell growth by targeting NRP1 (Liu et al, 2015), and inhibits liver cancer cell invasion through SMO (Huang et al, 2011). Like miR-133a and miR-199-2, miR-338-3p seems to be acting as a tumour suppressor by inhibiting cell proliferation.~~

For GO analysis, we generated a list of 3,001 genes with predicted miR-338-3p binding sites with TargetScan, yielding context+ scores between -0.04 and -1.48 (Supplementary Table ST87). g:Profiler identified only five biological process terms, three related to neuronal development and two to positive regulation of biological processes (Supplementary Fig. S4A, Table ST8). Panther identified additional terms like "embryo~~nie~~ development," while DAVID revealed terms such as "cartilage development," "fin morphogenesis," and "neural crest differentiation". No KEGG pathways were overrepresented. STRING analysis of genes with miR-338-3p binding sites and expression in NCC or derivatives identified *snai2*, *sox9b*, *kita*, and *pax2a* as central node genes (Fig. S3S4B).

~~These results suggest that miR-133a, miR-199-2, and miR-338-3p function as tumor suppressors, exhibiting reduced expression in various cancers. These miRNAs repress genes involved in EMT, migration, and cell proliferation—processes common to both cancer and NCC development. Although GO analysis revealed limited overrepresentation of NCC specific terms, with only a few associated with miR-133a and miR-338-3p, there was a notable bias toward neuronal development. However, these miRNAs may still regulate key genes critical for NCC development, such as *soxE*, *snai1/2*, and *msx1/2/3*.~~

miR-133a-3p, miR-199-23p, and miR-338-3p are differentially expressed during zebrafish embryogenesis.

Previous zebrafish miRNA profiling studies have reported developmental regulation of miR-133, miR-199 and miR-338 family members (Chen et al, 2005; Kloosterman et al, 2006). However, quantitative, stage-resolved expression data for the specific mature miRNAs (miR-133a-3p, miR-338-3p, and miR-199-3p), and for across the early stages examined here remain limited and vary across datasets. We therefore analyzed their temporal expression from 5 hpf onward using RT-qPCR on whole-embryo extracts. ~~Since the expression profiles of miR-133a, miR-199-2 and miR-338-3p in zebrafish remain incompletely characterized, we analyzed their temporal expression from the 5 hpf stage onwards using RT-qPCR.~~ Both miR-133a-3p and miR-199-2 3p showed the lowest expression at expression at 5 hpf among the analyzed stages (Fig. 3A–B). Their expression began to rise at 10 hpf, with a marked increase after 24 hpf. It should be noted that these profiles reflect global expression levels rather than NC-specific enrichment. Notably, this upregulation at 10 hpf aligns with the expression patterns of NCC genes that contain binding sites for these miRNAs, such as *sox9b* and *foxd3* (White et al, 2017; Moreno et al, 2022). Similarly, miR-338-3p was detected at 5 hpf— (right after zygotic genome activation (Vastenhouw et al, 2019)—), but its levels decreased between 10 and 24 hpf before

increasing again (Figure 3C). This pattern correlates with the downregulation of putative genes like *sox9b* and *snai1b* (White et al, 2017; Moreno et al, 2022). These findings suggest that these miRNAs exhibit temporally coordinated expression with their predicted target genes, highlighting their as potential role as regulators of gene expression during early zebrafish development.

To prioritize putative targets, we filtered genes present in at least two or all three predicted miRNA putative target gene lists. Given that miRNAs can simultaneously regulate multiple genes, and that each gene may be targeted by more than one miRNA, we filtered those genes present in at least two or all three predicted miRNAs putative target gene lists. Among these, only six NC-related genes were shared in all three lists, including *pdgfb*, *tbx20* and *ddx21* (see Table ST10 for NC-related genes, and Table ST11 for intersections Table ST9). Additionally, between 16 and 20 genes were common to two lists (Table ST119). Several of the identified target genes have been implicated in chondrocyte development, craniofacial structure formation, and melanophore differentiation. Given these associations, we decided to further investigate the roles of these miRNAs in these key biological processes.

To investigate the effects of miRNA gain-of-function, zebrafish embryos were microinjected with pri-miRNA constructs containing a tandem *dsRED* reporter sequence. We performed gain-of-function experiments in zebrafish embryos by microinjecting pri-miRNA constructs in tandem with a reporter *dsRED* sequence to increase the levels of miR-133a, miR-199-2, and miR-338-3p (dsRED-miR-133a, dsRED-miR-199-2, and dsRED-miR-338-3p, Fig. S4A-S5A). Control embryos received an mRNA encoding only the *dsRED* sequence (control-mRNA). As a control, embryos were injected with an mRNA generated from a construct containing only the *dsRED* sequence (dsRED-Control). We assessed during embryonic development both the presence of the reporter protein by fluorescence analysis and the pri-miRNA processing by stem-loop RT-qPCR. As expected, embryos collected at 24 hpf exhibited significant increases in miR-133a-3p, miR-199-3p, and miR-338-3p (hereafter called miR-133a, miR-199, and miR-338, respectively, Fig. S5B-D). As expected, significant increase in miR-133a, miR-199-2, and miR-338-3p levels were observed in RNA samples from microinjected embryos collected at 24 hpf (Fig. S4B-D). To ensure that the observed phenotypes were not secondary to general developmental delay or toxicity, we assessed gross morphology at 16 hpf and 48 hpf (Fig. S5E-Q). Quantification of somite number (16 hpf) and eye size (16 and 48 hpf) revealed no significant global developmental defects across conditions. These results confirm the efficacy of our overexpression strategy and establish a robust foundation for assessing the impact of these miRNAs on neural crest derivative development. These results provided a foundation for further studies on the impact of miRNA overexpression on of NC derivative development.

miR-133a and miR-338-3p overexpression alters melanophore development.

Zebrafish pigmentation is determined by three types of NC-derived chromatophores: melanophores (black pigment), iridophores (iridescent pigment that reflects light), and xanthophores (yellow pigment) (Kelsh et al, 1996). In this study, we specifically examined the development of melanophores, which become visible around 25 hpf in the otic vesicle region. These cells subsequently expand across the dorsal head region (Fig. 4A) and along the embryonic lateral line at 48 hpf (Fig. 4B-E) (Kelsh et al, 1996). To investigate the role of miR-133a and miR-338-3p, both having putative binding sites to genes involved in melanophore development (*dct*, *kita*, *sox10*, *tyrp1a*) (Greenhill et al, 2011), we performed a quantitative analysis on melanophore number present in 48 hpf larvae following the overexpression of each miRNA. Overexpression of either miRNA led to a reduction in the melanophore number located in the dorsal head region (Fig. 4C-B) and the lateral line (Fig. 4D-F). Given that these

miRNAs share putative target genes involved in melanophore development (like *dct*), we evaluated their potential synergistic effect through simultaneous overexpression. Co-overexpression of miR-133a and miR-338-3p produced a similar phenotype.

~~, suggesting that a maximal inhibitory effect on melanophore differentiation may already be achieved by either miRNA alone under the given experimental conditions. This aligns with the role of miRNAs as fine tuners of gene expression, where their regulatory impact does not necessarily scale linearly with increased expression. To explore the molecular basis of this phenotype, we analyzed the expression of *mitfa*, *kita*, *tyrp1a*, and *dct* (Greenhill et al, 2011). We further analysed the expression of *mitfa*, a master regulator of melanophore specification, and *dct* (dopachrome tautomerase), a key enzyme in melanin biosynthesis (Greenhill et al, 2011). Notably, *mitfa* lacks predictive binding sites for either miRNA, and its expression remained unchanged following miR-133a or miR-338-3p overexpression (Fig. 4E4C, Fig. S5S6D).~~

In contrast, *dct* contains two predicted binding sites for miR-338-3p and multiple predicted sites for miR-133a (Fig. S5S6E). Interestingly, while *dct* mRNA levels were unaffected in embryos overexpressing miR-338-3p, they were significantly reduced upon miR-133a overexpression (Fig. 4F4D). Both *kita* and *tyrp1a* contain one putative site for miR-338 (Fig. S6F-G), and their respective mRNA levels were reduced in miR-338-overexpressing embryos, but not in miR-133a-overexpressing embryos (Fig. 4G-H). For *kita*, this reduction was sustained upon combined overexpression. Altogether, these results show a complex and dynamic regulation, highlighting the redundancy typical of miRNA networks where multiple targets are shared. ~~This reduction may account for the observed decrease in pigmentation at 48 hpf in embryos microinjected with dsRED-miR-133a.~~

miR-133a and miR-338-3p overexpression adversely affects craniofacial cartilage development.

~~To evaluate cartilage formation, embryos microinjected with dsRED-miRNA constructs were developed until 5 days post-fertilization (dpf), then fixed and stained with Alcian Blue (Fig. 5A-B). Since all three miRNAs presented putative binding sites in genes related with chondrocyte development (*sox9b*, *runx3*, *col2a1b*, *colla2*) (Bell et al, 1997; Lefebvre et al, 1998; Daleq et al, 2012), we designed gain of function experiments to evaluate cartilage formation. Embryos microinjected with dsRED miRNA constructs were developed until 5 days post fertilization (dpf), then fixed and stained with Alcian Blue (Fig. 5A-B).~~ Overexpression of miR-199-2 did not produce any detectable changes in cartilage morphogenesis within the ~~analysed~~analyzed parameters (Fig. 5D, H-I). In contrast, miR-133a overexpression led to a significant reduction in Meckel's cartilage length and angle (ML and MAn, respectively; Fig. 5C, F-G), as well as a decrease ceratohyal angle (ChAn) and an increased ceratohyal-to-fin distance (ChD) (Fig. 5F-G). Similarly, larvae overexpressing miR-338-3p exhibited shorter palatoquadrate cartilages (PQ; Fig. 5E, J), reduced Meckel's cartilage angles, and an increased ceratohyal angle (ChAn; Fig. 5K).

Since miR-133a and miR-338-3p are predicted to co-target several genes associated with craniofacial development (like *sox9b*, *sox4a* and *wnt9b*) (Yan et al, 2005; Townley et al, 2008; Xu et al, 2018), we evaluated their potential synergistic effect. Our results revealed a significant reduction in both the Meckel's and ceratohyal angles, consistent with the trends observed when each miRNA was overexpressed individually (Fig. 6A-B). Additionally, co-microinjected larvae exhibited an increase in the Meckel's cartilage area (Fig. 6A-B). Interestingly, while the individual overexpression of dsRED-miR-133a or miR-338-3p led to

reduced lengths of the Meckel's (ML) and palatoquadrate (PQ) cartilages, and the distance from the fins to the ceratohyal cartilages (ChD), their dual overexpression restored these parameters to control levels. This suggests that the simultaneous presence of miR-133a and miR-338-3p, particularly in NC tissues where they are not typically co-expressed, may exert a complex regulatory influence on craniofacial development by modulating shared target genes. Alternatively, the restoration of ML values upon co-overexpression may indicate that each miRNA regulates distinct sets of genes with opposing effects on this parameter, ultimately balancing one another's impact. Further investigation is required to elucidate the precise molecular mechanisms underlying this interaction.

To further elucidate the molecular mechanisms underlying the observed craniofacial defects, we quantified the mRNA expression levels of *runx3*, *sox10*, and *sox9b*— (key regulators of cartilage development (Dalcq et al, 2012)—) using RT-qPCR. This analysis revealed a significant reduction in *runx3* expression in embryos microinjected with dsRED-miR-133a and dsRED-miR-338-3p, despite *runx3* containing only a single putative binding site for miR-133a (Fig. 6C, Fig. S65C). The effect of miR-338-3p on *runx3* expression may be indirect, potentially resulting from the dysregulation of upstream genes within the gene regulatory network (GRN) or a reduction in the specific cell population expressing *runx3*. Regarding *sox10*, mRNA levels were specifically reduced in dsRED-miR-133a microinjected embryos, consistent with the presence of a predicted binding site for this miRNA in the 3'UTR of *sox10* (Fig. 6D, Fig. S5A56A). Given that *sox10* is an early NC development marker (Lewis et al, 2004), its downregulation could contribute to the observed craniofacial abnormalities. Additionally, overexpression of both miR-133a and miR-338-3p resulted in decreased *sox9b* mRNA levels, likely due to the presence of multiple binding sites for these miRNAs in its 3'UTR (Fig. 6E, Fig. S5B56B). Since *sox9b* is a crucial regulator of chondrogenesis and NC-derived cartilage formation, its downregulation could be a key factor underlying the structural defects observed.

miR-133a and miR-338-3p bind to *sox9b* 3'UTR.

The observed reduction in *sox10*, *sox9b*, *det*, and *runx3* mRNA levels following the overexpression of miR-133a and miR-338-3p may be attributed to multiple mechanisms. These include a decrease in the specific cell populations expressing these genes, a reduction in the availability of transcription factors that activate their expression, or a direct interaction between the miRNA and the 3'UTR of the target gene. Such interactions can lead not only to mRNA degradation but also to translational repression (Kloosterman et al, 2006).

To determine whether these mRNA reductions stem from direct regulation by miR-133a or miR-338-3p, we performed a reporter assay using the *d4GFPn* gene (nuclear-localized destabilized eGFP, with a half-life of 4 hours (Sánchez-Vásquez et al, 2019)) to assess *sox9b* post-transcriptional translational repression. *Sox9b* was selected due to its critical role in cartilage development and the presence of multiple predictive miRNA binding sites in its 3'UTR. Briefly, we used two sensor constructs: *sox9b*-3'UTR-1, which contains a single putative binding site for miR-133a and another for miR-338, and *sox9b*-3'UTR-2, which harbors two putative binding sites exclusively for miR-338 (Fig. 7A, Fig. S6B). As a control, a third construct (*Xenopus* β -globin, control-3'UTR) was fused to the SV40-poly(A) sequence. The corresponding mRNAs were co-injected into 1-cell-stage embryos alongside dsRED-miR-133a, dsRED-miR-338, both, or control-mRNA, and GFP fluorescence was quantified at the 50%-epiboly stage (Fig. S6H). Briefly, we used different constructs in which the *d4GFPn* coding sequence fused to specific fragments of the *sox9b* 3'UTR. The first construct, *sox9b*-3'UTR-1, contains a single putative binding site for miR-133a and another for miR-338-3p,

while the second construct, *sox9b* 3'UTR-2, harbors two putative binding sites exclusively for miR-338-3p (Fig. 7G). The corresponding mRNAs were transcribed *in vitro* and co-injected into 1-cell stage embryos alongside dsRED-miR-133a, dsRED-miR-338-3p, both, or dsRED-control (Fig. S5F). Embryos were subsequently imaged at the 50% epiboly stage, and GFP fluorescence was quantified using QuantiFish. As an additional control, a third construct (control 3'UTR) was designed, in which *d4GFPn* was fused to the SV40 poly(A) sequence, ensuring that any observed effects were specifically attributed to *sox9b* 3'UTR sequences.

In embryos microinjected with *sox9b*-3'UTR-1, overexpression of both miR-133a and miR-338-3p significantly suppressed *d4GFPn* fluorescence translation, with the strongest effect observed upon co-microinjection of both miRNAs (Fig. 7B7C, Fig. S6J). Conversely, embryos injected with *sox9b*-3'UTR-2, exhibited a significant reduction in fluorescence only when dsRED-miR-338-3p was present (Fig. 7C7D, Fig. S6K). No significant changes were detected in embryos microinjected with the control-3'UTR, regardless of miRNA treatment (Fig. 7B, Fig. S6I Fig. 7A). These findings align with RT-qPCR results, reinforcing the notion that miR-133a and miR-338-3p directly regulate *sox9b* expression by targeting its 3'UTR. This regulation likely occurs through translational repression, mRNA degradation, or a combination of both mechanisms. The observed reduction in *sox9b* mRNA levels in RT-qPCR analyses may, at least in part, result from miRNA-induced mRNA destabilization

miR-338-3p overexpression results in higher NCC number.

The effects described thus far could stem from impaired cell differentiation, disruptions in migration patterns, and/or dysregulated NCC proliferation. To quantitatively evaluate changes in the NCC population, we microinjected Tg(*sox10*:GFP) embryos with dsRED-miR-133a, dsRED-miR-338-3p, or a control-mRNAs dsRED-control, followed by flow cytometry analysis to quantify GFP-positive cells (Fig. 8A). While miR-133a overexpression did not significantly alter NCC numbers, miR-338-3p overexpression resulted in a notable increase in NCC abundance (Fig. 8B). Interestingly, the co-expression of both miRNAs restored NCC numbers to control levels, suggesting a potential compensatory mechanism. The proliferation and differentiation of NCC are tightly regulated by numerous transcription factors and signalling pathways (Kelsh & Raible, 2002). Although miR-133a is predicted to target genes associated with cell proliferation, its overexpression did not significantly impact NCC numbers at 18 hpf. In contrast, miR-338-3p overexpression resulted in a marked increase in NCC abundance. This effect may stem from miR-338-3p's regulatory influence on genes involved in cell proliferation, such as *wif1* and *ddx21*, both of which have been implicated in NC expansion and developmental patterning (Yin et al, 2012a; Guglielmi et al, 2020; Johansson et al, 2020; Santoriello et al, 2020). The increase in NCC following dsRED-miR-338-3p overexpression may contribute to the defects in melanophore and chondrocyte development described above. Given miR-338-3p's involvement in both early (proliferation) and later (differentiation) stages of NC development, its dysregulation likely disrupts the balance between cell number and lineage specification, ultimately leading to the observed phenotypic abnormalities. In contrast, since miR-133a overexpression did not affect NCC proliferation, the associated defects in melanocyte and craniofacial development are more likely due to miR-133a mediated regulation of genes involved in NCC specification and differentiation, rather than alterations in cell number.

Discussion

MiRNAs are pivotal regulators of embryonic development and numerous diseases, including cancer, where they fine-tune gene expression at multiple levels (Kloosterman & Plasterk,

2006; Weiner, 2018). Due to their ability to target multiple mRNAs simultaneously (Kim, 2005) and considering the extensive network of transcription factors, enzymes, and structural proteins involved in the GRN (Gilbert, 2000; Martik & Bronner, 2017) of embryonic development, it is likely that many miRNA-mRNA interactions remain undiscovered. The parallels between NC development and cancer biology, particularly in processes such as EMT, migration, and invasion, suggest a shared regulatory landscape. Insights gained from one context can, therefore, enhance our understanding of the other.

Building on this concept, we selected miRNAs with established roles in tumorigenesis to investigate their regulatory influence on NC-derived tissues. Specifically, we focused on the roles of miR-133a-3p, miR-199-23p, and miR-338-3p, that are highly expressed in the premigratory NCC of *Danio rerio*. These miRNAs, which are conserved exclusively among vertebrates, have been implicated in cancer-related processes such as EMT, cytoskeletal reorganization, proliferation, migration, and apoptosis, as supported by both literature and bioinformatic analyses. Through this study, their contributions to NC-derived structures pathways were explored.

To validate the developmental relevance of these selected miRNAs, we examined their temporal profiles. While previous zebrafish profiling reported general developmental regulation of miR-133-3p and miR-199-3p families (Chen et al, 2005; Kloosterman et al, 2006), stage-resolved data for the specific isoforms examined here has been limited. Our analysis of whole-embryo lysates revealed that the upregulation of miR-133a-3p and miR-199-3p begins at 10 hpf, around gastrulation. Published zebrafish expression datasets report dynamic regulation of NCC regulators such as *sox9b* and *foxd3* across these stages (White et al, 2017; Moreno et al, 2022), providing context for potential temporal overlap between these miRNAs and predicted targets. Because these measurements were performed on whole embryos, the later increase likely reflects cumulative expression from multiple tissues (including muscle), whereas the early rise is at least compatible with roles during early NCC programs. Similarly, miR-338-3p was detected as early as 5 hpf (immediately following zygotic genome activation (Vastenhouw et al, 2019)) and displayed a biphasic pattern during early development. In published datasets, transcripts such as *sox9b* and *snai1b* are dynamically regulated across these stages (White et al, 2017; Moreno et al, 2022), which is consistent with the possibility of temporally coordinated miRNA-mRNA regulation during early embryogenesis. Overall, these observations support a potential role for miR-133a-3p, miR-199-3p, and miR-338-3p in modulating gene expression programs during early zebrafish development.

The literature suggests that miR-199-3p-2 plays a role regulating EMT and tumor cell migration, processes that are also crucial during early NC development (Giovannini et al, 2018; Wang et al, 2018; Zhang et al, 2019). However, no direct evidence has yet linked miR-199-2 3p to the regulation of specific NC derivatives. Notably, putative target genes of miR-199-2 3p include key NC specifiers such as *foxd3* (Martik & Bronner, 2017), as well as transcription factors involved in signaling pathways essential for NC specification, including Wnt and FGFs (Schmidt et al, 2013). The lack of phenotypic defects following miR-199-3p overexpression serves as an important internal control, demonstrating that the injection conditions did not induce non-specific toxicity or saturation of the RNAi machinery (Lund et al, 2011). While miR-199-2 3p overexpression did not result in detectable craniofacial defects, its role in NC development cannot be ruled out, particularly in processes such as cell migration and iridophore differentiation. This possibility is supported by the presence of putative binding sites in genes implicated in these processes, such as *ltk* (Petratou et al, 2018), as well as its

reported involvement in tumorigenesis and cell adhesion. Further studies, including functional assays and lineage tracing, are necessary to elucidate the precise role of miR-199-2-3p in NC development and its potential contribution to these biological processes.

The proper formation of craniofacial structures relies on the coordinated proliferation and positioning of cartilage progenitor cells (Yan et al, 2005). Transcription factors governing craniofacial development have been extensively characterized in *Danio rerio*, mice, and cultured cells. Our findings indicate that miR-133a-3p and miR-338-3p regulate the expression of *sox9b*, a critical transcription factor required not only for craniofacial cartilage formation but also for melanophore differentiation. Here, we have demonstrated that these miRNAs repress *sox9b* expression at the mRNA level (consistent with mRNA destabilization) and at the reporter protein level (consistent with translational repression and/or mRNA decay) by directly interacting with its 3'UTR. ~~at both the transcriptional and translational levels by directly interacting with its 3'UTR.~~ The observed reduction in *sox10*, *sox9b*, *dct*, and *runx3* mRNA levels could theoretically be attributed to indirect mechanisms, such as a decrease in the specific cell populations expressing these genes. However, our reporter assay results confirm that in the case of *sox9b*, miR-133a-3p and miR-338-3p exert their effect through a direct interaction with its 3'UTR. Since the reporter assay assesses protein levels (fluorescence) while our qPCR data showed reduced mRNA, this suggests the regulation involves both mRNA degradation and translational repression, consistent with established miRNA mechanisms (Kloosterman & Plasterk, 2006). Notably, co-overexpression of miR-133a and miR-338-3p exacerbates craniofacial malformations, suggesting a synergistic effect on cartilage development. This suggests that the simultaneous presence of miR-133a-3p and miR-338-3p may exert a complex regulatory influence, potentially by regulating distinct sets of genes with opposing effects on this parameter, ultimately balancing one another's impact. ~~This outcome likely results from their combined modulation of upstream regulators of chondrogenesis, highlighting their broader impact on NC-derived structures.~~

We also found that miR-133a-3p and miR-338-3p overexpression reduces expression levels of *runx3*, a crucial gene expressed in the pharyngeal endoderm that is essential for the activation of downstream cartilage GRN. ~~3p regulate *runx3*, a crucial gene expressed in the pharyngeal endoderm that is essential for the activation of downstream cartilage GRN.~~ The reduction in *runx3* expression following miR-133a-3p and miR-338-3p overexpression mimics phenotypes observed in *runx3* morphants, including defects in viscerocranial and anterior neurocranial cartilage (Dalcq et al, 2012). While miR-133a-3p likely directly targets *runx3*, miR-338-3p appears to act indirectly, potentially through the modulation of other transcription factors involved in chondrogenesis. These findings highlight the complexity of miRNA-mediated regulation, which can influence developmental pathways both directly and indirectly through highly interconnected GRNs. Furthermore, the specific reduction of *sox10* in miR-133a-3p-injected embryos likely contributes to these structural defects, as *sox10* is an early marker required for the specification of the NC populations that populate the pharyngeal arches (Lewis et al, 2004).

In melanogenesis, both miR-133a-3p and miR-338-3p also play significant regulatory roles. Vertebrate melanocytes (melanophores in fish) are critical for pigmentation, mating behaviors, individual recognition, and UV protection. They are also implicated in various pigmentation disorders, including albinism, vitiligo, and melanoma (Nordlund et al, 2007). Our data indicate that miR-133a-3p overexpression reduces *dct* expression, which may account for the observed decrease in melanocyte melanophore numbers. Interestingly, although *sox10* is required for the activation of *mitfa* (Greenhill et al, 2011), we did not detect a corresponding reduction in *mitfa*

mRNA levels at 24 hpf. Because *mitfa* lacks predicted binding sites for miR-133a-3p and miR-338-3p, the absence of a *mitfa* decrease likely reflects indirect buffering/compensatory regulation within the melanophore gene regulatory network and/or limited sensitivity of whole-embryo measurements at 24 hpf, given the restricted expression domain of *mitfa* relative to *sox10/sox9b* expression. This aligns with reports that zebrafish *mitfa* mutants retain a small population of pigmented cells that depend on *sox9b* (Greenhill et al, 2011). Since *sox9b* is transiently required in melanocyte precursors, the downregulation of *sox9b* by miR-133a-3p and miR-338-3p may disrupt early melanocyte activation, ultimately leading to reduced melanogenesis. The absence of detectable changes in the expression levels of *dct* (—an enzyme required for melanin biosynthesis (Greenhill et al, 2011)—) following miR-338-3p overexpression does not necessarily exclude miRNA-mediated regulation. It is possible that miR-338-3p modulates *dct* expression at the translational level rather than by inducing mRNA degradation, a mechanism consistent with the well-established dual role of miRNAs in post-transcriptional regulation. Further studies are needed to elucidate the precise molecular pathways through which these miRNAs influence zebrafish melanophore differentiation and pigmentation.

Contrary to its well-documented tumor-suppressor role (Huang et al, 2011, 2015; Liu et al, 2015), miR-338-3p overexpression led to an increased population of NC progenitor cells in zebrafish embryos. This expansion, coupled with the concomitant effect on melanophores and cartilage structures, suggests a significant block in lineage commitment. Mechanistically, this effect likely stems from miR-338-3p's regulatory influence on genes involved in cell proliferation, such as the predicted targets *wif1* and *ddx21*, both of which have been implicated in NC expansion and developmental patterning (Yin et al, 2012a; Guglielmi et al, 2020), alongside the downregulation of key pro-differentiation factors *sox9b* and *sox10*. By fine-tuning the *sox9b/sox10* regulatory axis, miR-338-3p likely maintains NCCs in a proliferative, undifferentiated state, preventing the transition from multipotent progenitors to specialized derivatives (Stevanovic et al, 2022). This unexpected outcome likely reflects the widespread effects of ubiquitous miRNA overexpression, which impacts all embryonic cells rather than being confined to NC tissues. The elevated proliferation of progenitor cells may contribute to developmental defects observed at later stages, highlighting the need for more refined spatial and temporal analyses to delineate the specific roles of miRNAs during development.

The dual role of miR-338-3p in promoting proliferation while inhibiting terminal differentiation echoes functions seen in tumorigenesis, where the subversion of differentiation programs is a hallmark of malignancy. Interestingly, co-expression of miR-133a-3p restored NCC numbers to control levels, suggesting a compensatory mechanism between the two miRNAs. In contrast, since miR-133a-3p overexpression did not affect NCC proliferation, the associated defects in melanophore and craniofacial development are more likely due to defects in specification and differentiation rather than alterations in progenitor cell number. This aligns with previous reports establishing miR-133a-3p as a key regulator of cell fate decisions and differentiation rather than proliferation in other developmental contexts (Ivey & Srivastava, 2010; Garcia-Padilla et al, 2022; Sharma et al, 2023).

While GO analyses did not reveal extensive enrichment of NC-related pathways, our findings strongly suggest that miR-133a and miR-338-3p play crucial roles in NCC development. These miRNAs likely exert their effects through both direct and indirect regulatory mechanisms, influencing key transcription factors such as *sox9b* and *runx3*. The interplay between these miRNAs and their target genes underscores their broader significance in cellular plasticity and differentiation.

It is important to note that bioinformatic predictions of miRNA-mRNA interactions often lack experimental validation. Demonstrating the co-expression of miRNAs and their targets within the same cell type is essential for confirming functional interactions (Xu et al, 2013; Yu et al, 2016; Sánchez-Vásquez et al, 2019; Lukiw & Pogue, 2020). Future studies employing *in situ* hybridization will be valuable for validating the co-localization of miRNAs and their putative targets, such as *sox10*, *runx3*, and *dct*. The intricate overlap of GRNs regulating various NC derivatives suggests that miRNA overexpression likely affects multiple tissues simultaneously. For instance, the dual regulatory effect of miR-133a and miR-338-3p on *sox9b* influences both craniofacial cartilage formation and melanogenesis, demonstrating how miRNAs orchestrate broad developmental processes.

In conclusion, our findings highlight the essential role of precise miR-133a-3p and miR-338-3p regulation in the development of NC-derived structures in *Danio rerio*, particularly craniofacial cartilage and melanophores. The craniofacial abnormalities observed in miRNA-overexpressing embryos likely stem from the direct repression of *sox9b* and *runx3* expression, while the reduction in the number of melanophore may result from downregulation of *dct*. Beyond their developmental roles, the shared regulatory features between NC formation and tumorigenesis underscore a broader function for these miRNAs in modulating cellular plasticity, proliferation, and differentiation. These insights not only deepen our understanding of miRNA-mediated control during vertebrate embryogenesis but also highlight the potential of miR-133a-3p and miR-338-3p as therapeutic targets in neurocristopathies and cancer. Future studies should further investigate their mechanistic roles and translational relevance in both developmental and pathological contexts.

Materials and methods

Zebrafish care

All zebrafish were handled in accordance with national and international guidelines (Westerfield, 2000). All protocols involving zebrafish from the Calcaterra Lab were approved in advance by the *Comité de Bioética para el Manejo y Uso de Animales de Laboratorio* of the Facultad de Ciencias Bioquímicas y Farmacéuticas, UNR (files 6060/374, resolution 207/2018). Adult zebrafish were maintained at a constant temperature of 27 ± 1 °C under a 14:10 h light/dark cycle (Westerfield, 2000). Wild-type (WT) fish from the AB strain (ZIRC, Oregon University, OR, USA) were used in this study. The transgenic lines used included Tg(*actb1*:eGFP), Tg(*sox10*:eGFP, *sox10*:mRFP) (Kwak et al, 2013) and Tg(*sox10*:eGFP).

Zebrafish embryos were obtained according to the methodology described by Kimmel et al (Kimmel et al, 1995). Briefly, selected adults were kept overnight at 28.5 °C in breeding tanks, with males and females separated in a 3-to-4 ratio, respectively. Embryos were staged by visualization under stereoscopic microscope (Olympus MVX10 stereoscopic microscope and Olympus C-60 ZOOM digital camera) based on morphological development in hours or days post-fertilization (hpf or dpf) at 28 °C (Kimmel et al, 1995).

Flow cytometry and FACS

Embryos at 16 and 28 hpf were manually dechorionated and deyolked using Ginsburg buffer (111.22 mM NaCl, 3.35 mM KCl, 2.7 mM CaCl₂, 2.38 mM NaHCO₃) through gentle pipetting and vortexing (600 rpm, 5 min). The samples were then centrifuged at 2500 rcf for 3 min at 4 °C, resuspended in 1X PBS containing 0.125% (w/v) Trypsin, and incubated at 37 °C for 15

min to facilitate dissociation. Following incubation, the samples were centrifuged (500 rcf, 15 min, 4 °C), resuspended in 500 μ L 1X PBS, filtered through a 40 μ m mesh, and counted.

To calibrate fluorescence-based cell sorting, wild-type and Tg(*actb1*:eGFP) lines were used as controls, representing 0% and 100% eGFP⁺ cells, respectively. For Fluorescence-Activated Cell Sorting (FACS) and Flow Cytometry cell counting, the transgenic lines Tg(*sox10*:eGFP, *sox10*:mRFP) and Tg(*sox10*:eGFP) were employed., respectively. FACS was conducted using BD FACSAria II and FACSAria III systems (BD Biosciences, NJ, USA) at a flow rate of 30 μ L/min. GFP⁺ cells were used to establish gating parameters for zebrafish cells (Fig. S1A) (Gallardo & Behra, 2013)s., while mRFP fluorescence served as a control, achieving >99% mRFP⁺ within the eGFP⁺ population. Cells sorted by FACS were directly collected in TRIzol (Invitrogen) for subsequent RNA extraction. Sorting efficiency was analyzed by measuring NCC marker *foxd3* and non-NCC marker *tbx5a* in each population (Fig. S1D).

Flow Cytometry cell counting was conducted using a BD Accuri C6 Plus system (BD Biosciences), with a flow rate of 30 μ L/min, ensuring a minimum of 100,000 events were recorded per condition. Each condition was analyzed in triplicate for each biological replicate. Data processing and analysis were performed using FlowJo software (BD Biosciences).

RNA-seq

Total RNA was extracted using TRIzol Reagent (Life Technologies) with DNase treatment to remove genomic contamination. RNA concentration and purity were assessed using the Qubit RNA BR Assay kit (Thermo Fisher Scientific), while RNA integrity was evaluated with the RNA 6000 Nano Bioanalyzer 2100 Assay (Agilent). After confirming RNA quality, samples from different experimental groups were used for library construction at each developmental stage.

Small RNA libraries were prepared using the TruSeq Small RNA Library Prep kit (Illumina) according to the manufacturer's protocol. Sequencing was performed on an Illumina NextSeq High Output platform in paired-end mode, generating 75 bp reads. Image analysis, base calling, and quality scoring were carried out using Illumina's Real-Time Analysis software (v3.4.4). The cleaned-up reads were checked using FASTQC and then aligned to the zebrafish genome using Bowtie (Langmead et al, 2009). Aligned reads were counted using HTSeq (Putri et al, 2022). Read counts for each sample included in the study were normalized after which the miRNA differential expression was analyzed using DESeq2 (Love et al, 2014). For plotting purposes, pseudocounts (\$\epsilon\$ ) were added to each normalized count value. The pseudocount was calculated as half of the minimum non-zero normalized count across all samples, resulting in a value of \$\epsilon = 0.0438\$. See Table ST2 and Fig. S1 for non-pseudocount normalized values.

Bioinformatics

Zebrafish (*Danio rerio*) pre-miRNA sequences were retrieved from Ensembl (Release: 98, Assembly: GRCz11) and miRBase (Version: 22.1). Putative miRNA target genes were identified using TargetScan Fish (targetscan.org/fish_62) (Ulitsky et al, 2012). Gene expression data for zebrafish were extracted from the EMBL-EBI Expression Atlas, based on experimental data from White *et al.*, 2017 (White et al, 2017; Moreno et al, 2022).

Gene Ontology (GO) and KEGG metabolic pathways analyses were conducted using multiple bioinformatic tools: DAVID (Version: v2022q4; GO and KEGG, FDR and p-value < 0.05) (Huang et al, 2009), PANTHER (Version: 10.5281/zenodo.4495804; GO, FDR and p-value <

0.05) (Mi et al, 2019), and g:Profiler (Version: e111_eg58_p18_f463989d; GO and KEGG, g:SCS Threshold and $p_{adj} < 0.05$) (Raudvere et al, 2019).

Statistical analyses were performed using Prism 9.5 (GraphPad Software, Boston, MA, USA). Depending on the experimental design, either a two-tailed t-test or ordinary/nested ANOVA followed by *post-hoc* Tukey's test was applied. Statistical significance was set at $p \leq 0.05$. Data visualization was performed with Prism 9.5 or with custom scripts using Python (v3.10), generated with pandas, seaborn and matplotlib libraries.

Reverse Transcription Followed by Quantitative PCR (RT-qPCR)

For RNA extraction, 35-45 embryos per condition were collected at the required developmental stage and rapidly flash-frozen in liquid nitrogen. Samples were either processed immediately or stored at -80 °C for long-term preservation. Embryos were homogenized in TRIzol, followed by chloroform extraction and isopropanol precipitation. Total RNA concentration was determined by measuring absorbance at 260 nm using a NanoVue spectrophotometer (GE Healthcare, Chicago, IL, USA).

Reverse transcription was performed using M-MLV reverse transcriptase (Promega, Madison, WI, USA) with 1 µg of RNA. Oligo-dT primers were used for mRNA reverse transcription, while specific stem-loop (Kramer, 2011) oligonucleotides were designed for each miRNA. Quantitative PCR (qPCR) was conducted using HOT FIREPol EvaGreen qPCR Mix Plus (Solis Biodyne, Tartu, Estonia) on a RealPlex4 thermocycler (Eppendorf, Hamburg, Germany). Zebrafish *rpl13* and *efl1a111* cDNAs were amplified as internal references. Primer sequences are provided in Table ST6. Data analysis was carried out using REST2009 software (Qiagen, Hilden, Germany) (Pfaffl et al, 2002), following MIQE guidelines (Bustin et al, 2009) to ensure experimental accuracy and reproducibility.

miRNA overexpression

For the overexpression experiments, the genomic regions encoding zebrafish *miR-133a* (miRBase Accession: MIMAT0001830; chr2: 4,113,889–4,114,465), *miR-199-2* (miRBase Accession: MIMAT0003155; chr5: 1,376,464–1,377,047), and *miR-338-3p* (miRBase Accession: MIMAT0048673; chr3: 51,907,303–51,907,638) were PCR-amplified and cloned into the pSP64T-dsRED expression vector using *EcoRI* and *XhoI* restriction sites (primer sequences provided in Supplementary Table S1, with restriction sites in lowercase).

For mRNA synthesis, plasmids were linearized with either *XbaI* or *BamHI* (Invitrogen, Carlsbad, CA, USA) and transcribed using the mMACHINE[®] SP6 transcription kit (Invitrogen). A ds-RED control vector lacking miRNA sequences was prepared as a negative control.

Microinjections were performed at the one-cell stage, targeting the yolk just beneath the blastomere, using a gas-driven microinjection system (MPPI-2 Pressure Injector, Applied Scientific Instrumentation, Eugene, OR, USA). Each embryo received 1.25 ng of transcript and was maintained at 28 °C until the required developmental stage for analysis.

Phenotype observation

Zebrafish embryos (~16 hpf) and larvae (48 hpf) from each treatment group were collected and immobilized in 3% (w/v) methylcellulose containing 0.15 mg/mL Tricaine (ethyl 3-aminobenzoate methanesulfonate salt; Sigma-Aldrich, A5040). Specimens were oriented

laterally and imaged using an Olympus MVX10 stereomicroscope equipped with an Olympus C-60 ZOOM digital camera. Somite number was counted manually. Eye surface area was quantified by outlining the eye perimeter using the freehand selection tool in FIJI (National Institutes of Health, Bethesda, MD, USA). Measurements were normalized to the mean of the control group.

Reporter Assay

The *sox9b* d4EGFPn-3'UTR reporter constructs were originally generated by (Steeman et al, 2021). An empty pSP64-T-d4EGFPn vector with *Xenopus* β -globin (*hbg1*, ENSXETG00000025667) 3'UTR was used as a control. Transcripts were microinjected into one-cell-stage embryos at a concentration of 1.5 ng, along with 1.25 ng of either dsRED-miR-133a, dsRED-miR-338, or control-dsRED (control-mRNA) transcripts. This study was repeated twice; each trial was conducted separately and individually involved the microinjection of more than 45 embryos per condition. ~~miR-133a-dsRED, miR-199-2-dsRED, miR-338-3p-dsRED, or control-dsRED transcripts.~~

Fluorescence from D4EGFPn d4EGFPn was assessed at the 50%-epiboly stage. At least 40 embryos per condition were imaged using an Olympus MVX10 stereomicroscope equipped with an Olympus C-60 ZOOM digital camera (Olympus, Tokyo, Japan). Fluorescence intensity was quantified using QuantiFish software (Stirling et al, 2020).

Alcian Blue Staining

To assess the effects on cranial structures, 30 larvae at 5 dpf were fixed overnight at 4 °C in 4% (w/v) paraformaldehyde prepared in 1X PBT (1X PBS with 0.1% (v/v) Tween-20). Following fixation, samples were washed four times with 1X PBT and stained according to previously established protocols (Weiner et al, 2019).

Images were captured using an Olympus MVX10 stereomicroscope equipped with an Olympus C-60 ZOOM digital camera. Cranial cartilage parameters were measured using ImageJ software (National Institute of Health, Bethesda, MD, USA) (Schneider et al, 2012) following previously reported methods (Weiner et al, 2020).

Melanophore counting

Larvae at 48 and 72 hpf were anesthetized using 0.15 mg/mL Tricaine (ethyl 3-aminobenzoate methanesulfonate salt, Sigma-Aldrich, Ref. A5040) and positioned in Petri dishes containing 3% (w/v) methylcellulose. Using two 30 G needles, larvae were carefully oriented to obtain lateral and dorsal images under an Olympus MVX10 stereomicroscope equipped with an Olympus C-60 ZOOM digital camera. Imaging was performed using top illumination against a white background. Pigmented cell counts were conducted manually by analyzing the acquiring images with Fiji software.

Figure Legends

Figure 1 – Differential expression of Neural Crest miRNAs. (A–B) Identification of differentially expressed miRNAs in NCC and non-NCC from 16 hpf (x-axis) and 28 hpf (y-axis) transgenic zebrafish samples. Over-represented miRNAs in either cell population are shown in yellow ($\log_2(\text{NCC}/\text{non-NCC}) > 2$). miR-145-5p, miR-133a-3p, miR-199-3p and miR-338-3p are highlighted in black, pink, green and blue, respectively. For plotting, pseudocounts (ϵ) were added to each normalized count value. See Table ST2 and Figure S1 for non-

pseudocount normalized values. The three miRNAs overrepresented in both timepoints are enclosed within a green dashed circle. Identification of differentially expressed miRNAs in NCC and non-NCC from 16 hpf (A) and 28 hpf (B) transgenic zebrafish samples. Overrepresented miRNAs in either cell population are shown in blue ($\log_2(\text{NCC}/\text{non-NCC}) > 2$). miR-145, miR-133a, miR-199-2 and miR-338-3p are highlighted in pink. (CB) Quantification of miR-145-5p, miR-133a-3p, miR-199-2-3p and miR-338-3p levels in NCC by RT-qPCR, compared to non-NCC; levels are, expressed as $\log_2(\text{NCC}/\text{non-NCC})$. Statistical analysis was performed using a two-tailed Student's t-test, * $p \leq 0.05$, ** $p \leq 0.01$, $n = 3$, mean \pm SEM. The circles correspond to the differential expression values obtained by the FACS/miRNA-seq experiment. Data information: Statistical analysis was performed using a two-tailed Student's t-test, * $p \leq 0.05$, ** $p \leq 0.01$, $n = 3$, mean \pm SEM.

Figure 2 – GO and STRING analysis of miR-133a-3p putative target genes. Similar analyses for miR-199-2 3p and miR-338-3p are presented in Supplementary Figures S2 S3 and S 34. (A) Selected overrepresented “Biological Processes” terms associated with miR-133a putative target genes, as identified by g:Profiler, Panther, and David/DAVID. (B) Selected miR-133a-3p putative target genes and their predicted interactions based on STRING. For the complete interaction network, refer to Supplementary-Figure S21.

Figure 3 – miRNA expression profiles during zebrafish embryonic development. (A–C) RT-qPCR quantification of miR-133a-3p (A), miR-199-3p2 (B) and miR-338-3p (C) levels from 5 to 48 hpf. Data information: Normalized to 5 hpf stage, $n = 3$, mean \pm SEM.

Figure 4 – Melanophore development in miR-133a and miR-338-3p overexpressing embryos. (A–B) Representative images of melanophores in the head (A) and quantification of melanophore numbers and lateral line (B) in the head of 48 hpf larvae microinjected with dsRED-miR-133a, dsRED-miR-338, both, or control-mRNA. The analyzed areas are highlighted in blue, with counted cells indicated by red arrowheads. Scale bar (A): 200 μm . regions. The analyzed areas are highlighted in blue, with counted cells indicated by red arrowheads. Scale bars: 200 μm . (C–D) mRNA expression levels of *mitfa* (C) and *dct* (D) in 24 hpf embryos microinjected with dsRED-miR-133a, dsRED-miR-338, both, or control-mRNA. (E–F) Representative images of melanophores (E), and quantification of melanophore numbers (F) in the lateral line of 48 hpf larvae microinjected with dsRED-miR-133a, dsRED-miR-338, both, or control-mRNA. The analyzed areas are highlighted in blue, with counted cells indicated by red arrowheads. Scale bar (E): 200 μm . (G–H) mRNA expression levels of *kita* (G) and *tyrp1a* (H) in 24 hpf embryos microinjected as described above. Statistical analysis (A–B, E–F): ANOVA followed by Tukey's post-hoc test; * $p \leq 0.01$, ** $p \leq 0.001$, $n = 25$, mean \pm SD. Statistical analysis (C–D, G–H): ANOVA followed by Tukey's post-hoc test; * $p \leq 0.05$, ** $p \leq 0.01$, $n = 3$, mean \pm SEM. Quantification of melanophore numbers in the head (C) and lateral line (D) of 48 hpf larvae microinjected with dsRED miR-133a, dsRED miR-338-3p, both, or dsRED control. Statistical analysis: ANOVA followed by Tukey's post hoc test; * $p \leq 0.01$, ** $p \leq 0.001$, $n = 25$, mean \pm SD. (E–F) mRNA expression levels of *mitfa* (E) and *dct* (F) in 24 hpf embryos microinjected with dsRED miR-133a, dsRED miR-338-3p, both, or dsRED control. Data information: Statistical analysis: ANOVA followed by Tukey's post hoc test; * $p \leq 0.05$, ** $p \leq 0.01$, $n = 2$, mean \pm SEM.

Figure 5 -Craniofacial development of zebrafish specimens overexpressing miR-133a, miR-199-2 and miR-338-3p. (A) Schematic representation of the craniofacial parameters analyzed: ML: Meckel's cartilage length; MAR: Meckel's cartilage area; ChL: Ceratohyal mean length; PQ: palatoquadrate mean length; CrD: cranial distance (from fins to Meckel's cartilage midpoint); ChD: ceratohyal distance (from fins to ceratohyal cartilage joint); MAn: Meckel's

cartilage angle; ChAn: ceratohyal angle. **(B–E)** Representative ventral view images of Alcian blue-stained cartilage in 5-dpf larva, with the head positioned to the left. Images show specimens developed from embryos microinjected with: ~~control-mRNA dsRED-control~~ **(B)**, dsRED-miR-133a **(C)**, dsRED-miR-199-2 **(D)** and dsRED-miR-338-3p **(E)**. **(F–K)** Craniofacial measurements larvae microinjected with: dsRED-miR-133a **(F–G)**, dsRED-miR-199-2 **(H–I)**, and dsRED-miR-338-3p **(J–K)**. **Data information:** Statistical analysis; Two-tailed Student's t-test; * $p \leq 0,05$, ** $p \leq 0,01$, *** $p \leq 0,001$, **** $p \leq 0,0001$, $n = 25$ for miR-199-2, $n = 22$ for miR-133a, $n = 48$ for miR-338-3p, mean \pm SD. Scale bar in **(B)**: 200 μ m.

Figure 6 – Synergistic Combined effect of miR-133a and miR-338-3p on craniofacial development. **(A–B)** Craniofacial measurements in larvae microinjected with dsRED-miR-133a, dsRED-miR-338-3p, both, or ~~control-mRNA dsRED-control~~. Statistical analysis: ANOVA followed by Tukey's post-hoc test; * $p \leq 0.05$, **** $p \leq 0.0001$, $n = 30$, mean \pm SD. **(C–E)** mRNA expression levels of *runx3* **(C)**, *sox10* **(D)** and *sox9b* **(E)** in 24 hpf embryos microinjected with dsRED-miR-133a, dsRED-miR-338-3p, both, or dsRED-control. **Data information:** Statistical analysis: ANOVA followed by Tukey's post-hoc test; * $p \leq 0.05$, $n = 2$, mean \pm SEM.

Figure 7 – Interaction between miR-133a, miR-338-3p, and the sox9b 3'UTR. **(A–C)** Schematic representation of the *sox9b* gene structure (green), highlighting the 3'UTR (empty box) and the predicted binding sites for miR-133a (blue) and miR-338 (pink). The *sox9b*-3'UTR-1 and *sox9b*-3'UTR-2 fragments, used in the d4GFPn reporter constructs, are indicated. **(B–D)** Quantification of d4GFPn fluorescence in embryos co-injected with either the ~~control-3'UTR (B), sox9b-3'UTR-1 (C), or sox9b-3'UTR-2 (D)~~ ~~control-3'UTR (A), sox9b-3'UTR-1 (B), or sox9b-3'UTR-2 (C)~~, alongside dsRED-miR-133a, dsRED-miR-338-3p, both, or control-mRNA or dsRED-control. **(D)** Schematic representation of the *sox9b* gene structure (green), highlighting the 3'UTR (empty box) and the predicted binding sites for miR-133a (blue) and miR-338-3p (pink). The *sox9b*-3'UTR-1 and *sox9b*-3'UTR-2 fragments, used in the d4GFPn reporter constructs are indicated. **Data information:** Statistical analysis was performed using ANOVA followed by Tukey's post-hoc test; * $p \leq 0.05$, ** $p \leq 0.01$, *** $p \leq 0.001$, **** $p \leq 0.0001$, $n = 45$, mean \pm SD.

Figure 8 – Flow cytometry-based quantification of NCCNCC proliferation. **(A)** Gating strategy used to identify zebrafish cells (~~Gallardo & Behra, 2013~~) (upper panel) and ~~distinguish~~ discrimination of GFP- negative and GFP-positive cell populations (lower panel). **(B)** ~~Flow Cytometry-based quantification of NCC-p~~ Proportion of NCCs in 18 hpf embryos microinjected with dsRED-miR-133a, dsRED-miR-338, both, or control-mRNA. ~~dsRED-miR-133a, dsRED-miR-338-3p, both, or dsRED-control.~~ **Data information:** Statistical analysis: ANOVA followed by post-hoc Tukey test, *** $p \leq 0.001$, $n = 5$, mean \pm SD.

Supplementary Material

Figure S1 – Differential expression of Neural Crest miRNAs. **(A)** Gating strategy used to sort zebrafish cells. The first plot is used to identify the zebrafish cells; the second plot is used to confirm singlet events; the third plot identifies GFP-negative (L) and GFP-positive (R) cells. The last plot shows the correlation between eGFP- and mRFP-positive cells from the Tg(*sox10*:eGFP, *sox10*:mRFP) zebrafish line. **(B–C)** Identification of differentially expressed miRNAs in NCC and non-NCC from 16 hpf **(B)** and 28 hpf **(C)** transgenic zebrafish samples. Over-represented miRNAs in either cell population are shown in yellow ($\log_2(\text{NCC}/\text{non-NCC}) > 2$). miR-145, miR-133a, miR-199 and miR-338 are highlighted in black, pink, green and blue, respectively. **(D)** Quantification of *foxd3* (NCC marker) and *tbx5a* (cardiomyocyte

marker, non-NCC) levels in NCC by RT-qPCR, compared to non-NCC; levels are expressed as $\log_2(\text{NCC}/\text{non-NCC})$.

Figure S1 S2 – STRING interaction network for miR-133a-3p putative target genes in zebrafish. The network depicts predicted and experimentally validated protein-protein interactions among genes containing putative miR-133a binding sites, as identified through bioinformatic analysis. Each node represents a protein encoded by the corresponding gene, while the connecting lines indicate interactions supported by multiple data sources.

Figure S2 S3 – STRING interaction network for miR-199-3p-2 putative target genes in zebrafish. (A) Selected overrepresented “Biological Processes” terms associated with miR-199-3p putative target genes, as identified by g:Profiler, Panther, and DAVID. (B) The network depicts predicted and experimentally validated protein-protein interactions among genes containing putative miR-199-2 binding sites, as identified through bioinformatic analysis. Each node represents a protein encoded by the corresponding gene, while the connecting lines indicate interactions supported by multiple data sources.

Figure S3 S4– STRING interaction network for miR-338-3p-3p t putative arget genes in zebrafish. (A) Selected overrepresented “Biological Processes” terms associated with miR-338-3p putative target genes, as identified by g:Profiler, Panther, and DAVID. (B) The network depicts predicted and experimentally validated protein-protein interactions among genes containing putative miR-338-3p binding sites, as identified through bioinformatic analysis. Each node represents a protein encoded by the corresponding gene, while the connecting lines indicate interactions supported by multiple data sources.

Figure S4 S5– miRNA overexpression strategy. (A) Schematic representation of the constructions used to generate the synthetic dsRED/miRNA mRNAs for overexpression experiments. The red region corresponds to the dsRED coding sequence, while the grey region represents the miRNA precursor sequence. These transcripts were synthesized using SP6 RNA polymerase (binding region indicated by arrows), and a poly-A tail was included in the construct to enhance mRNA stability and facilitate dsRED translation in zebrafish cells. (B–D) RT-qPCR quantification of miR-133a (B), miR-199-2 (C) and miR-338-3p (D) in RNA samples from 24 hpf zebrafish embryos microinjected with either dsRED-control or miRNA overexpression transcripts. (E–I) Representative images of early somitogenesis embryos microinjected with dsRED-control (E), dsRED-miR-133 (F), dsRED-miR-199 (G), dsRED-miR-338 (H), or dsRED-miR-133 and dsRED-miR-338 (I). (J–N) Representative images of early somitogenesis embryos microinjected with dsRED-control (J), dsRED-miR-133 (K), dsRED-miR-199 (L), dsRED-miR-338 (M), or dsRED-miR-133 and dsRED-miR-338 (N). Statistical analysis (B–D): Two-tailed Student’s t-test, $*p \leq 0.05$, mean \pm SEM. Statistical analysis (O–Q): ANOVA followed by Tukey’s post-hoc test, at least $n = 9$ (O, P), at least $n = 15$ (Q), mean \pm SD. ~~Data information: Statistical analysis was performed using a two-tailed Student’s t test, $*p \leq 0.05$, mean \pm SEM.~~

Figure S5 S6– miRNA putative binding sites. (A–EG) Schematic representation of the *Danio rerio* *sox10* (A), *sox9b* (B), *runx3* (C), *mitfa* (D), *dct* (E), *kita* (F), and *tyrp1a* (G) and *dct* (E) 3’UTR regions, highlighting the predicted binding sites for miR-133a (pink), miR-199-2 (green) and miR-338-3p (blue). The cloned fragments used in *sox9b* reporter assays are specifically highlighted in (B). (HF) Schematic overview of the reporter assay strategy. (I–K) Quantification of d4GFPn fluorescence in embryos co-injected with the control-3’UTR (I),

sox9b-3'UTR-1 (**J**), or sox9b-3'UTR-2 (**K**), alongside dsRED-miR-133a, dsRED-miR-338, both, or control-mRNA (Experiment 2). Statistical analysis was performed using ANOVA followed by Tukey's post-hoc test; * $p \leq 0.05$, ** $p \leq 0.01$, *** $p \leq 0.001$, **** $p \leq 0.0001$, $n = 45$, mean \pm SD.

Table ST1: Primer sequences for cloning and RT-qPCR.

Table ST2: miRNA RNA-seq, normalized counts for GFP+ 16hpf, GFP- 16hpf, GFP+ 28hpf and GFP- 28hpf.

Table ST3: miRNA RNA-seq, normalized counts for GFP+ 16hpf, GFP- 16hpf, GFP+ 28hpf and GFP- 28hpf.

Table ST4: List of zebrafish miR-133a putative regulated genes extracted from TargetScan.org and used for Gene Ontology analyses, with context+ score.

Table ST5: Gene Ontology results for miR-133a.

Table ST6: List of zebrafish miR-199-2 putative regulated genes extracted from TargetScan.org and used for Gene Ontology analyses, with context+ score.

Table ST7: Gene Ontology results for miR-199-2.

Table ST8: List of zebrafish miR-338-3p putative regulated genes extracted from TargetScan.org and used for Gene Ontology analyses, with context+ score.

Table ST9: Gene Ontology results for miR-338-3p.

Table ST10: Zebrafish NC-related genes from early segmentation stages.

Table ST11: miRNA NC gene lists intersection.

April 1, 2026

RE: Life Science Alliance Manuscript #LSA-2025-03431R

Dr. Tomás José Steeman
Instituto de Biología Molecular y Celular de Rosario
Ocampo y Esmeralda
Rosario 2000
Argentina

Dear Dr. Steeman,

Thank you for submitting your revised manuscript entitled "miR-133a-3p and miR-338-3p Shape Neural Crest Derivatives in Zebrafish". As you will see, both reviewers are satisfied with no further requests. We would be happy to publish your paper in Life Science Alliance pending final revisions necessary to meet our formatting guidelines.

MANUSCRIPT ORGANIZATION AND FORMATTING:

To avoid unnecessary delays in the acceptance and publication of your paper, please read the following information carefully. Full guidelines are available on our Instructions for Authors page, <https://www.life-science-alliance.org/authors>

- Please add the X and Bluesky handles of your host institute/organization, as well as your own, and/or one of the authors, in our system.
- Please label the tables as Table S1, Table S2, etc...

We welcome submissions of potential cover images for the issue of LSA in which your work would appear. If you have high quality images associated with this work, please feel free to email these, with a caption, to the journal office.

LSA encourages authors to provide a 30-60 second video where the study is briefly explained. We will use these videos on social media to promote the published paper and the presenting author (for examples, see <https://docs.google.com/document/d/1-UWCfbE4pGcDdcgzcmiuJl2XMBJnxKYeqRvLLrLSo8s/edit?usp=sharing>). Corresponding or first-authors are welcome to submit the video. Please submit only one video per manuscript. The video can be emailed to contact@life-science-alliance.org

FINAL FILES:

The following items are required for acceptance.

- An editable version of the final text (.DOC or .DOCX) is needed for copyediting (no PDFs).
- High-resolution figure, supplementary figure and video files uploaded as individual files: See our detailed guidelines for preparing your production-ready images, <https://www.life-science-alliance.org/authors>

The license to publish form must be signed before your manuscript can be sent to production. A link to the license to publish form will be available to the corresponding author only. Please take a moment to check your funder requirements.

Thank you for your attention to these final processing requirements. Please revise and format the manuscript and upload materials as soon as you are able.

Thank you for this interesting contribution to the literature. We look forward to publishing your paper in Life Science Alliance.

Sincerely,

Reviewer #1 (Comments to the Authors (Required)):

The authors have addressed most of the comments raised by me and another reviewer in the revised manuscript. Although some of the suggested additional experiments were not performed, the manuscript has been revised so that this limitation is made clear. I have no further comments at this stage.

Reviewer #2 (Comments to the Authors (Required)):

The authors have adequately addressed all my previous comments.

April 6, 2026

RE: Life Science Alliance Manuscript #LSA-2025-03431RR

Dr. Tomás José Steeman
Instituto de Biología Molecular y Celular de Rosario
Ocampo y Esmeralda
Rosario 2000
Argentina

Dear Dr. Steeman,

Thank you for submitting your Research Article entitled "miR-133a-3p and miR-338-3p Shape Neural Crest Derivatives in Zebrafish". It is a pleasure to let you know that your manuscript is now accepted for publication in Life Science Alliance. Congratulations on this interesting work.

Your article will publish open access upon publication under a CC-BY license.

DISTRIBUTION OF MATERIALS:

Again, congratulations on a very nice paper. I hope you found the review process to be constructive and are pleased with how the manuscript was handled editorially. We look forward to future exciting submissions from your lab.

Sincerely,
